# HydroFATE (v1): a high-resolution contaminant fate model for the global river system

**Heloisa Ehalt Macedo[1], Bernhard Lehner[1], Jim Nicell[2], and Günther Grill[1]**

[1]Department of Geography, McGill University, Montréal, QC H3A 0B9, Canada
[2]Department of Civil Engineering, McGill University, Montréal, QC H3A 0C3, Canada

**Correspondence:** Heloisa Ehalt Macedo (heloisa.ehaltmacedo@mail.mcgill.ca) and Bernhard Lehner (bernhard.lehner@mcgill.ca)

**Abstract.** TS1 Pharmaceuticals and household chemicals are neither fully consumed nor fully metabolized when routinely used by humans, thereby resulting in the emission of residues down household drains and into wastewater collection systems. Since treatment systems cannot entirely remove these substances from wastewaters, the contaminants from many households connected to sewer systems are continually released into surface waters. Furthermore, diffuse contributions of wastewaters from populations that are not connected to treatment systems can directly (i.e., through surface runoff) or indirectly (i.e., through soils and groundwater) contribute to contaminant concentrations in rivers and lakes. The unplanned and unmonitored release of such contaminants can pose important risks to aquatic ecosystems and ultimately human health. In this work, the contaminant fate model HydroFATE is presented, which is designed to estimate the surface-water concentrations of domestically used substances for virtually any river in the world. The emission of compounds is calculated based on per capita consumption rates and population density. A global database of wastewater treatment plants is used to separate the effluent pathways from populations into treated and untreated and to incorporate the contaminant pathways into the river network. The transport in the river system is simulated while accounting for processes of environmental decay in streams and in lakes. To serve as a preliminary performance evaluation and proof of concept of the model, the antibiotic sulfamethoxazole (SMX) was chosen, due to its widespread use and the availability of input and validation data. The comparison of modelled concentrations against a compilation of reported SMX measurements in surface waters revealed reasonable results despite inherent model uncertainties. A total of 409 000 km of rivers were predicted to have SMX concentrations that exceed environmental risk thresholds. Given the high spatial resolution of predictions, HydroFATE is particularly useful as a screening tool to identify areas of potentially elevated contaminant exposure and to guide where local monitoring and mitigation strategies should be prioritized.

## 1 Introduction

Contaminants of emerging concern (CECs) are deemed to be an important source of risk due to their potential adverse environmental impacts in the global water system (Gavrilescu et al., 2015; Noguera-Oviedo and Aga, 2016). For instance, pharmaceutically active compounds such as analgesics, antibiotics, estrogens, and antiepileptics, which are in widespread use globally, are not fully metabolized by the human body; thus, after their excretion and subsequent delivery into the wastewater collection and treatment system, they may ultimately reach the aquatic environment (Aydin et al., 2019; Kümmerer, 2009; Palli et al., 2019; Patrolecco et al., 2018; Praveena et al., 2018). The ongoing release of these compounds and other household chemicals through wastewater discharges often has unknown or poorly understood effects on the environment and human health. Importantly, most wastewater treatment plants (WWTPs) are not specifically designed to remove these contaminants before discharging effluents into receiving waterbodies, such as rivers, lakes, or oceans (Rizzo et al., 2019). As such, wastewaters that are collected from domestic sources and de-

livered via sewer systems to a WWTP may be only partially – or not at all – treated for such substances, thereby causing the WWTP to serve as a concentrated point source of contamination of CECs into aquatic ecosystems (Daughton and Ternes, 1999; Petrie et al., 2015; Meyer et al., 2019). In addition to these point sources, diffuse sources of contaminants from populations who are not connected to the sewage system can add to the pollution of waterbodies (Lapworth et al., 2012). Risks associated with these contaminants are further exacerbated due to the limited monitoring of their presence in wastewaters and receiving waterbodies into which they are discharged and incomplete assessment of their impacts downstream. In turn, this lack of information leads to poor regulatory oversight to safeguard the health of aquatic ecosystems and that of populations that rely on them as a source of water (Daughton, 2014). Moreover, robust estimates of current and future changes in water quality are needed to achieve sustainable management of water resources to ensure clean and accessible water for all, as promoted by Sustainable Development Goal (SDG) 6 (Strokal et al., 2019; Tang et al., 2019; van Vliet et al., 2019).

When measurements of waterborne contaminants are unavailable or insufficient to make informed decisions regarding water pollution arising from CECs, simulation models can be used instead to represent the hydrodynamic and water quality conditions of the waterbody. Contaminant fate models (CFMs), also known as environmental exposure models or georeferenced river models, focus on instream processes such as transport and degradation after the compounds' release from point and non-point sources. CFMs are specifically designed to predict realistic distributions of contaminants in a river catchment (Aldekoa et al., 2016). Examples of models operating at regional to global scales include GREAT-ER (Aldekoa et al., 2013; Feijtel et al., 1997), LF2000-WQX (Johnson et al., 2007), GIS-ROUT (Wang et al., 2000), PhATE (Anderson et al., 2004), Mike 11 (Havnø et al., 1995), WorldQual (Voß et al., 2012), ePiE (Oldenkamp et al., 2018), and GWAVA (Johnson et al., 2013). These models require information about the hydrological characteristics of the catchment, consumption rates of the chemical substances, and fate parameters that describe their instream decay. These requirements can limit the performance of the models in regions where this information is unreliable or scarce (Grill et al., 2016).

Water pollution caused by CECs is an issue of global concern, and water quality assessments must therefore be spatially consistent and comparable across the world to be able to identify locations of high contaminant concentration and regional trends in water pollution over time at a global scale. One of the challenges for global contaminant fate modelling is the lack of spatial consistency in datasets for model inputs, especially in regions where data are insufficient to support detailed assessments (Kroeze et al., 2016; Strokal et al., 2019; Tang et al., 2019). For this reason, only a few global-scale CFMs exist, and those that do are typically limited to

certain substances and relatively coarse spatial resolutions. For example, GLOBAL-FATE (Font et al., 2019), which was created as an open-source down-the-drain model that includes lake and reservoir modules as well as wastewater input information at a global scale, operates at a 7 km spatial grid. The Global TCS model (van Wijnen et al., 2018) was created to simulate the transport of the antibacterial agent triclosan in global rivers at a 0.5° spatial resolution (i.e., corresponding to approximately 55 km grid cells at the Equator).

To our knowledge, all currently existing global CFMs that require the quantification of the load of wastewater into the river system use population density and national sanitation statistics as proxies to derive the necessary input data (e.g., Beusen et al., 2015; Font et al., 2019; Hofstra et al., 2013; Mayorga et al., 2010; Strokal et al., 2019; Van Drecht et al., 2009; van Puijenbroek et al., 2019; Williams et al., 2012). More specifically, calculations are based on the fraction of the population connected to sewage systems per country. The main source of these statistics is the World Health Organization and the United Nation Children's Fund (WHO/UNICEF) Joint Monitoring Program (JMP) for Water Supply, Sanitation and Hygiene (WASH), which provides regular global reports on drinking-water and sanitation coverage for tracking progress toward SDG 6 (WHO and UNICEF, 2021). This dataset allows for differentiation of wastewater treatment services between countries and over time, but it does not account for spatial variability inside national boundaries, except for an assumed correlation with population density. Herrera (2019) also points out several discrepancies between national-level data and JMP-WASH data. In addition, the dataset does not contain specific locations of wastewater discharge, which can have important implications with respect to the distribution of contaminants in the river system.

Another important limitation of existing global water quality models is that they do not account for diffuse sources of pollution arising from populations who are not connected to WWTPs or for the natural attenuation of contaminants that occurs along their pathway from a source in the landscape through the soil or subsurface before reaching a waterbody. The contribution of diffuse pollution can be substantial as revealed by the high aquatic concentrations of pharmaceuticals that have been measured in countries with low rates of sanitation (Hanna et al., 2020; K'Oreje et al., 2012; Khan et al., 2013).

Grill et al. (2016, 2018) introduced a regional CFM that estimates the emission of household contaminants and their subsequent transport in river networks at high spatial resolution (river network derived from 500 m grid cells). In this model, transport in the river system is simulated using the global river routing model HydroROUT (Lehner and Grill, 2013). It has been applied and evaluated with respect to its ability to model the fate of several pharmaceuticals in the Saint Lawrence River basin in Canada (Grill et al., 2016), the pharmaceutical diclofenac in India (Shakya, 2017), and human hormones in China (Grill et al., 2018). These assess-

ments included not only WWTPs as point sources but also accounted for diffuse sources of contamination from populations not served by WWTPs while accounting for natural attenuation.

In the present work, the CFM by Grill et al. (2016, 2018) is fully developed to operate at a global scale in order to (1) serve as a large-scale screening tool for assessing CECs from domestic sources, especially as a precursor for potential risk assessments; (2) predict critical locations in river networks of potentially high aquatic contaminations; and (3) inform the development and implementation of guidelines, regulations, and mitigation strategies that aim to limit chemical pollution and safeguard human and ecosystem health. The model enhancement and expansion are performed by integrating a global WWTP database (HydroWASTE; Ehalt Macedo et al., 2022) and by distinguishing the pathways of contaminants from their population source to the river network depending on whether they are treated (i.e., either in centralized WWTPs or in decentralized facilities) or untreated (i.e., either from urban or rural diffuse sources). The capability of this global model, hereafter called HydroFATE, is then evaluated by applying it to estimate the global distribution of the antibiotic sulfamethoxazole (SMX) in the river network and by comparing the resulting predictions of environmental concentrations to field measurements reported in the literature. SMX was selected for this proof-of-concept case study due to the abundance of SMX field measurements in surface waters reported globally and the broader availability of model input parameters in the literature compared to many other CECs.

Given the broad goals, the main focus of the model development presented herein is to predict spatial variations in contaminant exposure and to achieve a level of model performance where estimates of concentrations in the river network are mostly within an order of magnitude of reported field measurements, which is generally considered adequate for these types of screening models (Johnson et al., 2008; Oldenkamp et al., 2018). HydroFATE, with its inherent global applicability due to its reliance on pre-existing data in addition to its high spatial resolution, aims to provide a tool for scientists, practitioners, and regulators to advance and focus their work, especially in regions where data are lacking.

## 2 Data

### 2.1 River and lake network

The various raster and vector layers representing the river network and catchment boundaries in HydroFATE were obtained from the global hydrographic database HydroSHEDS (Lehner et al., 2008), which was derived from digital elevation data provided by NASA's Shuttle Radar Topography Mission (SRTM) at 90 m (3 arcsec) resolution. For the present study, we used a derivative of this database in vector format, termed RiverATLAS (Linke et al., 2019), which was extracted at 500 m (15 arcsec) grid cell resolution and represents all rivers and streams where the average discharge exceeds $100 \, \mathrm{L \, s^{-1}}$ or the upstream catchment area exceeds $10 \, \mathrm{km^2}$, or both. The resulting global river network comprises 8 477 883 individual river reaches with an average length of 4.2 km, representing a total of $35.8 \times 10^6 \, \mathrm{km}$ of rivers. Each river reach has an associated contributing catchment with an average area of $15.7 \, \mathrm{km^2}$.

Every river reach in RiverATLAS is provided with a series of precalculated hydro-environmental characteristics. From this database, we used the long-term (i.e., 1971 to 2000) average naturalized river discharge in our study. The discharge estimates were derived from the global hydrological model WaterGAP version 2.2 (Müller Schmied et al., 2014), which were downscaled from their original resolution of 0.5° grid cells to the RiverATLAS resolution of 500 m using geostatistical techniques (Lehner and Grill, 2013). In addition to annual average discharge estimates, minimum discharges (i.e., the lowest monthly flow value within an average year) were also used for assessments under low-flow conditions.

To account for lake processes, a global database called HydroLAKES was used that provides the shoreline polygons of 1.4 million lakes with a surface area of at least 10 ha (Messager et al., 2016). All lakes in HydroLAKES are associated with RiverATLAS via their lake pour points.

### 2.2 WWTP information

HydroFATE incorporates the locations and characteristics of wastewater treatment plants (WWTPs) as provided by the HydroWASTE database (Ehalt Macedo et al., 2022). This database contains information on 58 502 WWTPs and provides details for each on the actual location of the plant, the estimated outfall location, and attributes that are relevant for the purposes of this study (including population served, treated-wastewater discharge, and level of treatment, i.e., primary treatment, such as solids removal through mechanical cleaning and sedimentation; secondary treatment, which includes biological processes; and advanced (tertiary or higher) treatment through extra filtration or chemical treatment). HydroWASTE was developed by combining regional and national WWTP datasets and adding auxiliary information, including Open Street Map data, global population data, and the high-resolution river network from RiverATLAS which was used to georeference WWTP outfall locations.

With respect to its implementation in HydroFATE, of the 58 502 WWTPs in the database, the following were excluded (note that some records fall into more than one category): (1) 1682 WWTPs that were labelled as *closed, non-operational, decommissioned, projected, proposed*, or *under construction*; (2) 379 WWTPs that have their outfall location outside of any catchment that is associated with the river network of RiverATLAS (e.g., small islands); (3) 199

WWTPs that serve a population of zero according to records; and (4) 9521 WWTPs that have their outfall location within 10 km from the ocean coast. The latter category was excluded to avoid overestimation of contaminant loads in coastal rivers as, given the locational uncertainties in HydroWASTE of up to 10 km (Ehalt Macedo et al., 2022), effluents from WWTPs with estimated outfall locations near the coast might, in reality, discharge directly into the ocean. Of the remaining 47 547 WWTPs, some share their original location inside the same 500 m pixel (i.e., the resolution of the HydroFATE model) and thus were aggregated to the final number of 46 270 point sources of wastewater discharge into the global river network.

To account for small or decentralized wastewater treatment systems (DWTSs) not included in the HydroWASTE database, such as septic tanks, HydroFATE uses country-level statistics provided by the JMP-WASH program (WHO and UNICEF, 2021). For the purposes of our study, sanitation data for each country were acquired for the year 2015, and the information termed "Proportion of population using improved sanitation facilities (wastewater treated)" was selected.

### 2.3 Population and urban area grids

Global gridded population distributions of the year 2015 were provided by the WorldPop dataset (WorldPop and CIESIN, 2018), which was produced using a combination of census, geospatial, and remotely sensed data in a spatial modelling framework (Tatem, 2017). The WorldPop data were disaggregated from their original spatial resolution of 1 km to the same resolution as the HydroFATE model (500 m) to allow for spatially consistent calculations.

Information on the location of global urban areas is determined in HydroFATE according to the Global Human Settlement (GHS) database (Pesaresi and Freire, 2016) for the year 2015. The global information provided by GHS was used to calculate the attribute "urban extent" in RiverATLAS, and it is based on fine-scale satellite imagery, census data, and volunteered geographic information. GHS data were disaggregated from their original spatial resolution of 1 km to 500 m.

### 3 Methodology

The regional CFM previously developed by Grill et al. (2016, 2018) simulates both the emission of household contaminants and their subsequent transport towards and within the river system. Building on this earlier work, we here enhance and then expand this CFM, termed HydroFATE, to the global scale. Figure 1 provides a conceptual representation of the HydroFATE model. Contaminant emissions are determined based on population distribution, per capita consumption of the modelled substance, human metabolism, and wastewater treatment removal, or natural attenuation, depending on the

pathway from the source to the waterbody. Emissions from populations served by a WWTP or by smaller and decentralized wastewater treatment systems (DWTSs) are reduced in proportion to the treatment efficiency, which is based on the level of treatment, i.e., primary, secondary, or advanced (tertiary or higher), that is provided by the WWTP. Emissions arising from populations that are not served by any type of wastewater treatment system are attenuated by a direct discharge coefficient (ddc) depending on the distance from the river network and whether the emission is located in a rural or urban area (Grill et al., 2018). The combined loads from all pathways of contaminants inside the catchment boundaries of an individual river reach are aggregated as the total local contaminant load of the reach. HydroFATE then employs the generic river routing model HydroROUT (Grill et al., 2014, 2019; Lehner and Grill, 2013) to simulate the transport of the chemical substance in the river system, accumulating the contaminant load downstream and accounting for instream decay and removal in lakes. Finally, the predicted environmental concentration (PEC) for every river reach is calculated by dividing the sum of the local total contaminant load plus the incoming load from upstream reaches by the long-term river discharge of the reach.

The methodologies used to simulate the amount of contaminant emissions and the routing of contaminant loads along rivers and through lakes were previously described at the regional scale (Grill et al., 2016, 2018). While these basic processes do not change when applied at the global scale, the model was expanded in the present study by incorporating novel global-scale input data. Furthermore, the model was enhanced by introducing a spatially explicit differentiation of various wastewater and contaminant pathways depending on the access of global populations to wastewater treatment (see dotted rectangle in Fig. 1). Within each pathway, contaminants are removed following different removal efficiencies offered by treatment facilities or different levels of natural attenuation in the soil and subsurface.

### 3.1 Determination of contaminant pathways

HydroFATE calculates contaminant emissions using contaminant-specific information (i.e., the annual per capita consumption and the excretion fraction) and the number of people connected to the river system. This connection occurs through different pathways depending on the sanitation system at the location in question. Using the global WWTP database HydroWASTE (Ehalt Macedo et al., 2022), a population grid, an urban extent grid, and additional sanitation data, six types of contaminant pathways from populations into the river network were determined and incorporated into the HydroFATE model (see Fig. 1). These are point sources of treated wastewater from populations connected to WWTPs that provide (1) a primary level of treatment, (2) a secondary level of treatment, or (3) an advanced (i.e., tertiary or higher) level of treatment; (4) decentralized sources

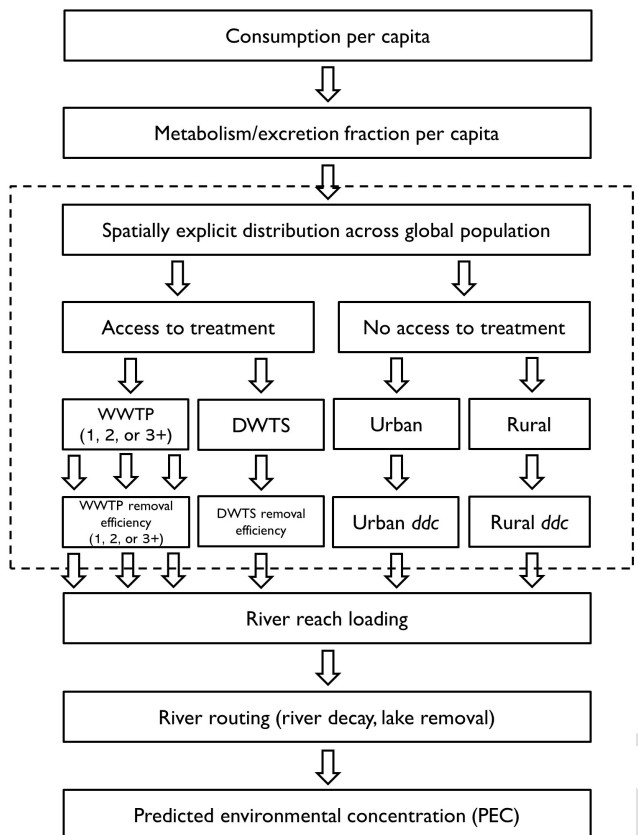

**Figure 1.** Conceptual representation of the contaminant fate model HydroFATE. The abbreviations "1, 2, or 3+" refer to the level of treatment of each WWTP as primary, secondary, or tertiary/advanced, respectively. The abbreviation "ddc" refers to the direct discharge coefficient. The dotted rectangle highlights steps involving contaminant pathways, i.e., processes developed or enhanced in the present study. See text for details and model description. Modified from Grill et al. (2018). TS2

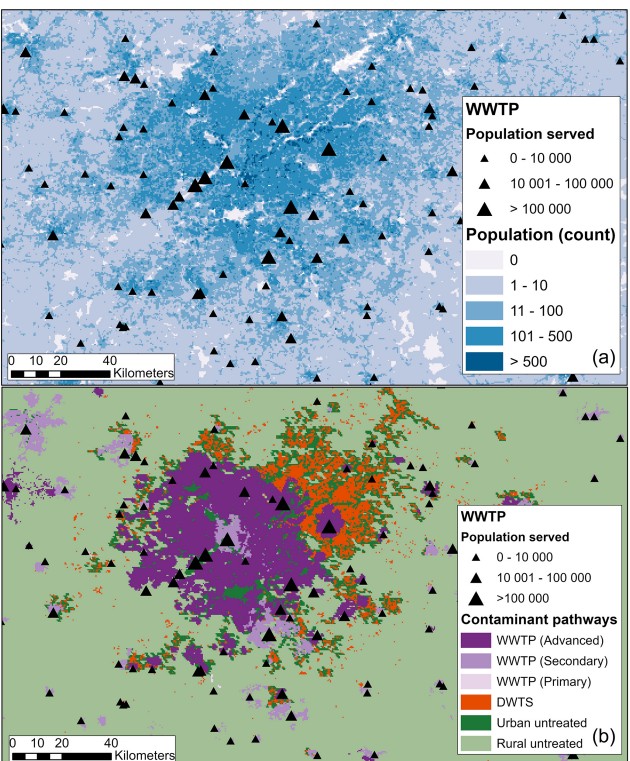

**Figure 2.** Example of the location of WWTPs and the population distribution **(a)** and the modelled contaminant pathway allocation **(b)** for the Atlanta metropolitan area in the United States. The areas shaded in purple **(b)** show the estimated service areas associated with individual WWTPs (black triangles). The populations residing in these areas are connected to the river network as point sources based on the discharge locations of their respective WWTPs. The populations residing in the orange areas **(b)** are identified as being associated with decentralized wastewater systems (DWTSs) and are connected to the river network as diffuse but treated sources within the catchment of each river reach. The populations residing in the areas represented by shades of green **(b)** are associated with untreated wastewater contributions and are connected to the river network as diffuse sources within the catchment of each river reach.

of treated wastewater from populations not connected to a WWTP but served by DWTSs; (5) diffuse sources of untreated wastewater from populations in urban areas; and (6) diffuse sources in rural areas. The methods described in more detail below assign a contaminant pathway for every pixel in a global population grid. Figure 2 illustrates an example of the resulting pathway allocation in comparison to the population distribution at a metropolitan area and its rural surroundings.

First, populations are allocated to individual WWTPs. Although HydroWASTE provides details on the number of people served by a WWTP, it does not specify the spatial distribution of the population served nor the service area associated with it; that is, it does not provide explicit information that is required to spatially allocate the populations that are served versus those not served by WWTPs (top panel of Fig. 2). The service area of a WWTP depends on several local factors not easily obtainable at the global scale,

such as decisions of the administrative unit responsible for the facility and the distribution of underground pipes that transfer the wastewater to the facility. Studies have presented different approaches to associate the area contributing to a WWTP. For instance, Keller et al. (2006) defined it as the nearest upstream contiguous urban area from the WWTP discharge point within 2 km, estimating the population served by the WWTP based on the number of people in this contributing area. However, the largest WWTP in their study served only 32 000 people (expressed as population equivalent), whereas HydroWASTE contains almost 5000 WWTPs that serve more than 100 000 people. Kapo et al. (2017) and Grill et al. (2018) associated the WWTP service area to an administrative unit, but these studies were developed in coun-

tries where the information on administrative units is widely available (i.e., USA and China, respectively), which is not typically the case at a global scale.

To allocate explicit spatial population distributions to individual WWTP locations in HydroFATE, we developed a method that follows the approach of Shakya (2017). This approach assumes that a WWTP can serve populations both upstream and downstream as wastewater can be pumped and directed in complex underground sewage systems. It also assumes that the service area of a WWTP can exceed the nearest contiguous urban area, with larger WWTPs typically serving larger distances and populations. Shakya (2017) tested different buffer sizes (i.e., from 5 to 30 km at 5 km increments) in India to determine the best-fit service area for different WWTP sizes by comparing the population within the buffer to the reported number of population served. Since distribution and characteristics of WWTPs in different regions of the world can vary substantially, we expanded upon this approach by using an iterative process instead of predefined buffer sizes.

The final WWTP allocation method assigns populations from the WorldPop population grid (WorldPop and CIESIN, 2018; see Sect. 2.3) to the point locations of WWTPs using a ranking system as described in detail in Appendix A. The method considers the distance of each population pixel from the WWTP, the size of the population served by the WWTP, whether a population pixel is categorized as "urban" or not, and whether candidate pixels are clustered in contiguous areas. The settings and thresholds applied in this method were initially set to those reported by Shakya (2017) and were then refined and finalized in a successive trial-and-error approach in which intermediate results were mapped, visually inspected for plausibility, and statistically tested to verify whether they led to further improvements. The final allocation procedure assigns population pixels to individual WWTPs until the reported total of served population of each WWTP is reached or until maximum distance thresholds are exceeded. Once the allocation is completed, the contaminant pathway from each allocated population pixel to the river reach is defined by the WWTP discharge location and can be separated into one of three treatment levels (primary, secondary, or tertiary/advanced) as specified in HydroWASTE (purple colors in bottom panel of Fig. 2).

Besides explicit WWTP pathways, HydroFATE also accounts for sources of potential contamination from decentralized wastewater treatment systems (DWTSs) that are not included in HydroWASTE, such as household septic tanks. To this end, for every country, the difference was calculated between the aggregated population served by WWTPs according to HydroWASTE and according to the global database on sanitation JMP-WASH (WHO and UNICEF, 2021). If JMP-WASH reported higher numbers of population served, this difference was assigned successively to the pixels with highest population numbers within the respective country borders that have not been allocated to WWTPs. The wastewater

pathway type of these pixels thus defaults to that of DWTSs (orange color in bottom panel of Fig. 2) and includes a specific removal efficiency. In the absence of explicit information, it is assumed that after DWTS treatment the effluent discharge directly enters the surface drainage system at the pixel's location within a catchment and then flows to the catchment's associated river reach.

Finally, all remaining population pixels that were not assigned in any of the previous steps were considered to be diffuse wastewater sources and were classified as "untreated" (green colors in bottom panel of Fig. 2). They were separated between rural and urban using an urban area grid (see Sect. 2.3). All population pixels classified as diffuse sources thus have a defined contaminant pathway that goes from the pixel's location within a catchment to the catchment's associated river reach. The contaminant removal along this pathway in soils and the subsurface is determined through distinct urban vs. rural attenuation functions.

### 3.2 Incorporation of contaminant pathways into HydroFATE

The results of the various population allocation steps described above are used as inputs into the HydroFATE model. The total input of contaminants from treated pathways into each river reach is the sum of the contributions from all WWTPs (i.e., point sources) releasing wastewater into that reach and the contribution from populations served by DWTS (i.e., decentralized sources):

$$
L_{t,r} = \left( \sum_i^r \left( P_{\mathrm{WWTP},i} \times \left( 1 - \frac{e_{\mathrm{WWTP},j}}{100\%} \right) \right) \right.
$$
$$
\left. + \left( \sum_m^c P_{\mathrm{DWTS},m} \times \left( 1 - \frac{e_{\mathrm{DWTS}}}{100\%} \right) \right) \right) \times L_{\mathrm{cap}}, \tag{1}
$$

where $L_{t,r}$ is the total load of the contaminant in river reach $r$ originating from treated pathways in the reach catchment $c$ contributing to $r$ (g d$^{-1}$); $P_{\mathrm{WWTP},i}$ is the population (persons) served by each WWTP $i$ connected to river reach $r$; $P_{\mathrm{DWTS},m}$ is the population (persons) served by DWTS from pixel $m$ inside catchment $c$; $L_{\mathrm{cap}}$ is the per capita load (excreted) of the contaminant (g cap$^{-1}$ d$^{-1}$); and $e_{\mathrm{WWTP},j}$ and $e_{\mathrm{DWTS}}$ are the removal efficiencies (%) of WWTPs and DWTS, respectively, releasing wastewater into the reach at treatment level $j$ (primary, secondary, or advanced).

To estimate the diffuse contributions from populations in urban and rural areas that are not served by wastewater treatment systems, it is assumed that not all human releases of untreated wastewater enter directly into surface waterbodies. This is due to various processes of natural attenuation such as absorption in soils or deposition in land surface depressions (Lapworth et al., 2012). Unfortunately, the factors affecting the natural attenuation and partial release of effluents are currently not well understood. Therefore, a proxy variable termed the direct discharge coefficient (ddc) is incorpo-

rated into the model to represent the fraction (dimensionless) of contaminant load from untreated pathways that reaches a waterbody after processes of natural attenuation. For example, for baseline model applications (see Sect. 4), the direct discharge coefficient for urban populations was set to 0.8 and for rural populations to 0.5, respectively, following Grill et al. (2018). The higher coefficient value for urban areas is due to the presence of impervious surfaces leading to more direct disposal of wastewater to nearby rivers and streams. While these methods are simplistic in comparison to real soil processes, no previous large-scale model considers untreated pathways as sources of contaminants, which can be substantial in regions with limited treatment infrastructure. The total input of contaminants from untreated pathways to each river reach is then calculated as TS3

$$
L_{u,r} = \left( \left( \sum_m^c P_{\text{urb},m} \times \text{ddc}_{\text{urb}} \right) \right.
$$
$$
\left. + \left( \sum_m^c P_{\text{rur},m} \times \text{ddc}_{\text{rur}} \times F_m \right) \right) \times L_{\text{cap}}, \qquad (2)
$$

where $L_{u,r}$ is the total load of the contaminant arriving at reach $r$ from all untreated pixels $m$ inside the reach catchment $c$ (g d$^{-1}$); $L_{\text{cap}}$ is the per capita load (excreted) of the contaminant (g cap$^{-1}$ d$^{-1}$); $P_{\text{urb},m}$ and $P_{\text{rur},m}$ are the total count of population (persons) following the untreated pathway from pixel $m$, in urban and rural areas, respectively; and ddc$_{\text{urb}}$ and ddc$_{\text{rur}}$ (dimensionless) are the direct discharge coefficients representing the proportion of contaminant load from untreated pathways that are discharged into the river reach $r$ from urban and rural areas, respectively. $F_m$ (dimensionless) is a factor by which loads from rural populations are additionally reduced based on an inverse distance relationship that accounts for limited connectivity in areas that are further away from the river network, following the approach by Grill et al. (2018):

$$
F_m = \left( D_{m,r} + 1 \right)^{-1}, \qquad (3)
$$

where $F_m$ (dimensionless) represents the fractional distance-based contribution factor for pixel $m$; and $D_{m,r}$ (kilometres) is the Euclidean distance between pixel $m$ and river reach $r$. This equation delivers fractional contribution values between 0 and 1, with 1 for locations closest to the river, 0.5 at a distance of 1 km, and continuously decreasing values as the distance increases. In contrast to the original method used in Grill et al. (2018), we refrained from normalizing the factor (i.e., by constraining $F_m$ to 0 at the furthest distance in each reach catchment), considering that contaminant contributions from any distance can reach the river system. Also, we used Euclidean distances rather than distances along the surface hydrological flow path (as proposed by Grill et al., 2018) assuming that contaminants can also travel through soils and groundwater. We tested the sensitivity of the parameter settings by doubling and halving both the distance value and

the exponent in Eq. (3), finding that the resulting uncertainty ranges were below those of other model parameter settings.

### 3.3 River and lake routing

The mass transport in HydroFATE follows a "plug-flow" approach (Pistocchi et al., 2010). That is, a "plug" of substance mass is accumulated downstream as the sum of the input from the current and all upstream reaches flowing into the current reach (Grill et al., 2018):

$$
L_{a,r} = \left( L_{t,r} + L_{u,r} + \sum_n L_n \right) \times d_{s,r} \times d_{l,r}, \qquad (4)
$$

where $L_{a,r}$ represents the total load of the contaminant accumulated at the end of river reach $r$ (g d$^{-1}$), calculated as the mass influx from treated pathways ($L_{t,r}$) plus the mass influx from untreated pathways ($L_{u,r}$) plus the total load (after decay) from those upstream reaches $n$ ($\sum_n L_n$) that directly discharge into reach $r$, multiplied by the instream decay factor $d_{s,r}$ (dimensionless) and the lake decay factor $d_{l,r}$ (dimensionless) that apply at reach $r$. The instream degradation of a chemical substance in the river body, if applicable, is expected to decrease at a rate proportional to its mass and is calculated assuming first-order decay:

$$
d_{s,r} = e^{-kt_r}, \qquad (5)
$$

where $d_{s,r}$ (dimensionless) is the instream decay factor for reach $r$, $t_r$ is the time a plug of water needs to travel through the river reach $r$ (days), and $k$ is a first-order decay constant specific to the contaminant (d$^{-1}$) which determines the rate of environmental decay in the river (Grill et al., 2018). Note that the inverse of $k$ represents the half-life of the chemical in the environment.

As an important partial contaminant sink, lakes were integrated and modelled as "completely stirred reactors" (CSTRs) (Anderson et al., 2004). The degradation of a chemical substance in lakes within the river network is calculated as follows:

$$
d_{l,r} = \frac{Q_r}{Q_r + (k \times V_r)}, \qquad (6)
$$

where $d_{l,r}$ (dimensionless) is the lake decay factor for reach $r$; $Q_r$ is the river discharge (L d$^{-1}$) at the river reach $r$; $k$ is the first-order decay constant specific to the contaminant (d$^{-1}$); and $V_r$ is the combined volume (L) of all lakes along river reach $r$. The locations and characteristics of lakes in HydroFATE are derived from the global lake database HydroLAKES (Messager et al., 2016). If there are no lakes in the river reach, $d_{l,r}$ is equal to 1.

To calculate the predicted environmental concentration (PEC$_r$) of the contaminant at river reach $r$ (ng L$^{-1}$), the final contaminant load (after any accumulation or removal) at

reach $r$ ($L_{a,r}$; g d$^{-1}$) is divided by the river discharge ($Q_r$; L d$^{-1}$) at the same location:

$$PEC_r = \frac{L_{a,r}}{Q_r} \times \frac{10^9 \, ng}{1 \, g}. \tag{7}$$

## 4  Model application and performance evaluation: concentration of sulfamethoxazole in the global river network

To evaluate the global applicability and general performance of the HydroFATE model, including the new distinction into six contaminant pathways as described in Sect. 3.2, the model was used in a proof-of-concept study to predict the distribution of the antibiotic sulfamethoxazole (SMX) in the global river network. SMX is considered a contaminant of emerging concern (Wilkinson et al., 2022), and it was selected due to a relatively high level of data availability, including global per capita consumption and metabolism fraction, WWTP removal efficiency, instream decay constants, and measured environmental concentrations (MECs) reported for numerous rivers and streams around the world. The PECs in surface waters were calculated based on long-term naturalized discharge as provided for all river reaches in the RiverATLAS database (Linke et al., 2019). The resulting PECs were compared to MECs to evaluate the model's predictive ability. Furthermore, PECs were also used to assess the exposure associated with SMX based on a comparison of PECs in surface waters relative to the reported predicted no-effect concentration (PNEC), which is the concentration threshold below which no adverse effects of exposure are observed in laboratory-based toxicity tests (Archundia et al., 2018; Hernando et al., 2006).

Finally, to further assess the model's performance under a range of alternative conditions, HydroFATE was run for a total of four scenarios based on plausible ranges of configuration settings and parameters extracted from literature sources.

### 4.1  Input data

#### 4.1.1  SMX properties

Sulfamethoxazole (SMX) is a sulfonamide antibiotic, usually sold in combination with trimethoprim. When consumed, SMX is rapidly absorbed upon oral administration, with metabolism mainly hepatic (Rudy and Senkowski, 1973). Residues are mostly excreted in urine, and the proportion of unchanged substance can be between 10 % and 30 %, depending on urine pH (Straub, 2016). Based on a comprehensive literature search, Straub (2016) found 36 publications that reported 190 removal efficiencies of SMX in WWTPs, with an average of 21 % removal, a median of 49 %, and an interquartile range from 2 % to 73 %. Archundia et al. (2018) compiled six different studies that measured environmental

decay in rivers, finding an average first-order decay constant of 0.73 d$^{-1}$, a median of 0.13 d$^{-1}$, a minimum of 0.034 d$^{-1}$, and a maximum of 2.88 d$^{-1}$.

For the main mode run in this study, the average excretion fraction and the median values of wastewater removal efficiency and instream decay constant were used (see Table 1, baseline scenario). But we also explored the ranges of possible values and how they affect the model outputs (see Table 1, alternative scenarios; for more details, see Sect. 4.2.1. below). Since Straub (2016) does not provide specific removal efficiencies for SMX for different treatment levels of WWTPs, the same removal efficiency was assumed for primary, secondary, and advanced treatment levels.

#### 4.1.2  SMX global consumption

Country-level averages of annual consumption per capita of SMX were necessary to estimate emissions to the river network. Klein et al. (2018) analyzed and estimated the consumption of various antibiotics in the world based on the IQVIA database, which reports annual sales for the period of 2012 to 2015 for 91 countries. For our study, the annual consumption of SMX for each country was assumed to be the average between the four available years as provided by Klein et al. (2018). For countries not included in the IQVIA database, the consumption rate was extrapolated based on the average per capita SMX use from the same income group (World Bank, 2019) following the methodology described by Klein et al. (2018).

#### 4.1.3  Measured environmental concentrations (MECs)

To evaluate the predictive ability of HydroFATE, 227 data points of MECs were compiled from literature sources (see Fig. 3 for their location and see Sect. S1 in the Supplement for literature sources). MECs are reported from every continent except Oceania. The average of reported values is 390 ng L$^{-1}$, the median is 28 ng L$^{-1}$, the minimum is 0.23 ng L$^{-1}$, and the maximum is 21 000 ng L$^{-1}$. In addition, 134 non-detects (i.e., concentrations reported were lower than could be detected based on the analytical method used to measure SMX in water samples) were compiled. In order to be selected for inclusion as a MEC in the model evaluation, the literature source must have reported the specific location (i.e., in the form of coordinates, river names, or river intersections) of the measurements. In addition, we discarded any MEC where the literature source explicitly mentioned that the dominant use of antibiotics in the catchment feeding the river was associated with veterinary or industrial activities, since the current version of HydroFATE is not adapted to account for these sources. Most MECs reported do not include corresponding comprehensive information on the river characteristics, such as width or average discharge, which would have helped to identify the precise location in the river network, or on the conditions under which the mea-

surement was taken, such as actual discharge amount or flow season (i.e., low flow, average, or high flow). Therefore, the actual location and amount of the reported MEC may not always accurately correspond to the referenced river network location and/or the assumed flow conditions in HydroFATE.

### 4.1.4 Predicted no-effect concentration (PNEC)

Since reports of SMX detection in rivers and streams first began to appear, its potential impact on the environment and human health has been assessed in several ways. That is, for exposure assessments, PNEC values often serve as thresholds to evaluate the level of environmental exposure to contaminants (Hernando et al., 2006). There are two values of PNEC for SMX published to date in literature. First, the PNEC minimum inhibitory concentration (PNEC-MIC), for which a value of $16\,000\,\text{ng}\,\text{L}^{-1}$ was estimated by Bengtsson-Palme and Larsson (2016), is intended to be protective of antibiotic resistance. Second, the PNEC environment (PNEC-ENV), for which a value of $600\,\text{ng}\,\text{L}^{-1}$ was estimated by Ferrari et al. (2004), is based on ecotoxicology data and is intended to protect aquatic function. For the purposes of the present case study, the lower value of PNEC ($600\,\text{ng}\,\text{L}^{-1}$) was selected to protect against any possible impact in the exposure analysis.

### 4.2 Model application

#### 4.2.1 Scenarios

A total of four main scenarios were created to portray plausible settings of parameters and model configuration, as outlined in Table 1. Scenario 1 represents baseline conditions using input parameters and model configurations that are expected to yield the most plausible predictions for average-flow conditions based on reported values in literature as described in Sect. 4.1.1 and in Grill et al. (2018) for the direct discharge coefficients. Scenario 2 represents low-flow conditions, but it otherwise maintains the same parameters and configuration settings as Scenario 1. Scenarios 3 and 4 represent low-end and high-end settings yet still within plausible ranges. That is, Scenario 3 (conservative case) uses parameter and configuration settings that represent minimum load emissions and maximum removal efficiencies (including full substance removal in lakes) to represent low-end contaminant concentrations in the river network, and vice versa for Scenario 4. In the absence of relevant literature values, plausible boundaries for the direct discharge coefficients of Scenarios 3 and 4 were set slightly above 0 (i.e., representing complete decay along untreated pathways) and below 1 (i.e., representing no decay along untreated pathways). Additional scenarios (see Table S2 in the Supplement) were designed for the model performance evaluation to analyze the individual contributions of selected parameters and model configu-

rations on the output, including a worst-case scenario assuming that no removal processes affect the contaminant load.

#### 4.2.2 Exposure assessment

The ratio of PEC to PNEC was used as an indicator to designate levels of SMX in the global river network that can lead to environmental health concerns. For that purpose, risk quotients, $\text{RQ}_{r,S}$ (dimensionless), were calculated for every river reach $r$ and every scenario configuration $S$ using the value of PNEC for SMX of $600\,\text{ng}\,\text{L}^{-1}$ and the calculated $\text{PEC}_{r,S}$ ($\text{ng}\,\text{L}^{-1}$) at every reach $r$ for the respective scenario $S$:

$$\text{RQ}_{r,S} = \frac{\text{PEC}_{r,S}}{\text{PNEC}}. \tag{8}$$

In instances where the risk quotient is greater than or equal to 1 ($\text{RQ}_{r,S} \geq 1$), it is assumed that this exposure level can cause negative environmental impacts (Archundia et al., 2018; Hernando et al., 2006).

#### 4.2.3 Performance evaluation

The performance of the model was evaluated by comparing reported MECs of SMX (see Sect. 4.1.3) and PECs calculated using HydroFATE at the coinciding river reaches. The goodness-of-fit indicators used to quantify model performance included the normalized root mean square error (NRMSE), the percentage of bias (PBIAS), the Nash–Sutcliffe efficiency coefficient (NSE; Nash and Sutcliffe, 1970), and the Kling–Gupta efficiency coefficient (KGE; Gupta et al., 2009). In addition to the 227 MECs, 134 measurements were classified as "not detected" or "not quantified". To evaluate these cases, PECs at the same locations were verified to determine if they were correctly predicted to be below the detection or quantification limit (LOD or LOQ, respectively), depending on detection limits reported in the respective studies. Besides the baseline calculations of Scenario 1, Scenarios 3 and 4 were evaluated as they benchmark plausible low-end and high-end variations of parameter and configuration settings, and Scenarios 5–7 (including 14 subscenarios; see Sect. S2 in the Supplement) were developed to specifically test the uncertainty ranges introduced by individual parameter and configuration settings.

### 4.3 Case study results

#### 4.3.1 Global emission of SMX to rivers

The global consumption of SMX from the world's population is estimated at $2.6 \times 10^6\,\text{kg}\,\text{yr}^{-1}$. From these, $2.4 \times 10^6\,\text{kg}\,\text{yr}^{-1}$ are consumed by populations with emission pathways that can potentially reach the river and lake system (including processes of metabolism, excretion, treatment, and/or natural attenuation), while the remaining $0.2 \times 10^6\,\text{kg}\,\text{yr}^{-1}$ are consumed by populations with direct emission pathways to the ocean. For the baseline scenario (Ta-

**Table 1.** Scenarios designed to represent plausible parameter and model configuration settings to simulate the global distribution of SMX in rivers. Excretion fraction is the fraction of the consumed amount of SMX that is excreted after metabolism. Wastewater treatment removal efficiency is the percentage of SMX that is removed in treatment facilities (WWTPs or DWTSs). For other parameter and configuration settings, see the main text.

| | | Parameter settings | | | | Configuration settings | |
|---|---|---|---|---|---|---|---|
| Scenario | Excretion | Wastewater treatment | Direct discharge coefficient | | Instream decay | Lake removal | Discharge |
| | fraction TS4 | removal efficiency (%) | Urban $ddc_{urb}$ | Rural $ddc_{rur}$ | constant $k$ ($d^{-1}$) | | condition |
| 1   Baseline, average flow | 0.2 | 49 | 0.8 | 0.5 | 0.13 | CSTR removal | Average flow |
| 2   Baseline, low flow | 0.2 | 49 | 0.8 | 0.5 | 0.13 | CSTR removal | Low flow |
| 3   Low-end case, average flow | 0.1 | 73 | 0.2 | 0.2 | 2.88 | Full removal | Average flow |
| 4   High-end case, low flow | 0.3 | 2 | 0.9 | 0.9 | 0.03 | No removal | Low flow |

ble 1), a total of 220 000 kg yr$^{-1}$ of SMX (9 % of global consumption) is estimated to reach rivers and lakes (Table 2). From this, 38 % is from pathways with some form of wastewater treatment (WWTP or DWTS) versus 62 % from untreated pathways. The results show that although most of the consumption occurs among rural populations without access to treatment (i.e., 44 % of total consumption), natural attenuation, as simulated in this study, has a high potential to remove the substance in rural areas before it reaches the rivers (i.e., resulting in only 26 % of total emission to rivers and lakes). Populations in urban areas without access to wastewater facilities were modelled to have similar emissions as populations with access to treatment (i.e., 36 % versus 38 % of total emissions, respectively). The processes simulated in this study that are responsible for removing portions of the substance along its way from the consumer to the final destination at the ocean or an inland sink are, in order of quantity removed, metabolism (80 % of total consumption is removed), natural attenuation in rural areas (6.5 %), wastewater treatment in WWTPs and DWTS (3.6 %), instream decay (3.5 %), lake removal (1.7 %), and natural attenuation in urban areas (0.7 %). The total SMX reaching the ocean or an inland sink through rivers amounts to 94 100 kg yr$^{-1}$ (4.0 % of global consumption).

Table 2 shows the 20 countries that are estimated to have the largest emissions of SMX. India accounts for the highest national emission to rivers and lakes (32 300 kg yr$^{-1}$), despite its lower-than-average per capita consumption (801 µg d$^{-1}$). This is due to a combination of large populations and a lack of access to wastewater treatment in urban areas. South Africa shows the highest per capita consumption (6220 µg d$^{-1}$), while China shows one of the lowest (67 µg d$^{-1}$), but it is still among the top 20 emitters.

Overall, the spatial patterns of contaminant emissions to rivers and lakes are very similar to the global patterns of consumption, with an average emission to consumption ratio of 9.2 % (Table 2). In Ethiopia, the ratio of emission to consumption was predicted to be on the low end (7.3 %), which is mostly due to the population being predominantly rural without access to treatment (77.9 %), contributing to

52.5 % of the total emission of SMX. In contrast, other countries such as Indonesia, Vietnam, and Egypt have a ratio above the global average (11.5 %, 11.4 %, and 11.2 %, respectively) due to the main source of SMX being untreated urban pathways (i.e., impervious surfaces with less attenuation) or treated pathways (i.e., wastewater treatment removal efficiency for SMX is lower than the assumed proportion of SMX removed by processes of natural attenuation in rural areas).

Figure 3 illustrates the resulting spatial distribution of SMX concentrations in the global river network for the two baseline scenarios: Scenario 1 corresponding to average-flow conditions and Scenario 2 corresponding to low-flow conditions. Generally, higher concentrations are predicted for rivers in countries with high emissions, such as India, United States, Pakistan, and South Africa. Nonetheless, even in countries that are not among the highest emitters, such as many African countries, low river discharges can cause high concentrations of contaminants in the rivers, especially during low-flow conditions.

### 4.3.2   Exposure assessment

The results of baseline Scenario 2 (low-flow conditions) predict aquatic exposure to SMX concentrations above the PNEC (i.e., risk quotient $\geq 1$) for 1.7 % (i.e., 409 000 km) of all rivers in the world with long-term annual average discharge above 0.1 m$^3$ s$^{-1}$ (Table 3). India, Pakistan, and Sudan show the largest extents of rivers in this category which indicates a potential risk for environmental health. This percentage decreases to 0.1 % (corresponding to 29 000 km) for baseline Scenario 1, i.e., when average-flow conditions are assumed. Pakistan has a particularly high percentage of rivers in the risk category for both scenarios (i.e., 33.5 % and 8.6 % of all rivers for Scenarios 2 and 1, respectively).

To assess the contribution of instream decay processes (i.e., decay in rivers and removal in lakes) to the reduction in contaminant concentrations, the increase in length of rivers with SMX concentrations exceeding PNEC was calculated assuming that these processes are not taking effect (i.e., the respective first-order decay constant $k$ is set to 0). Globally,

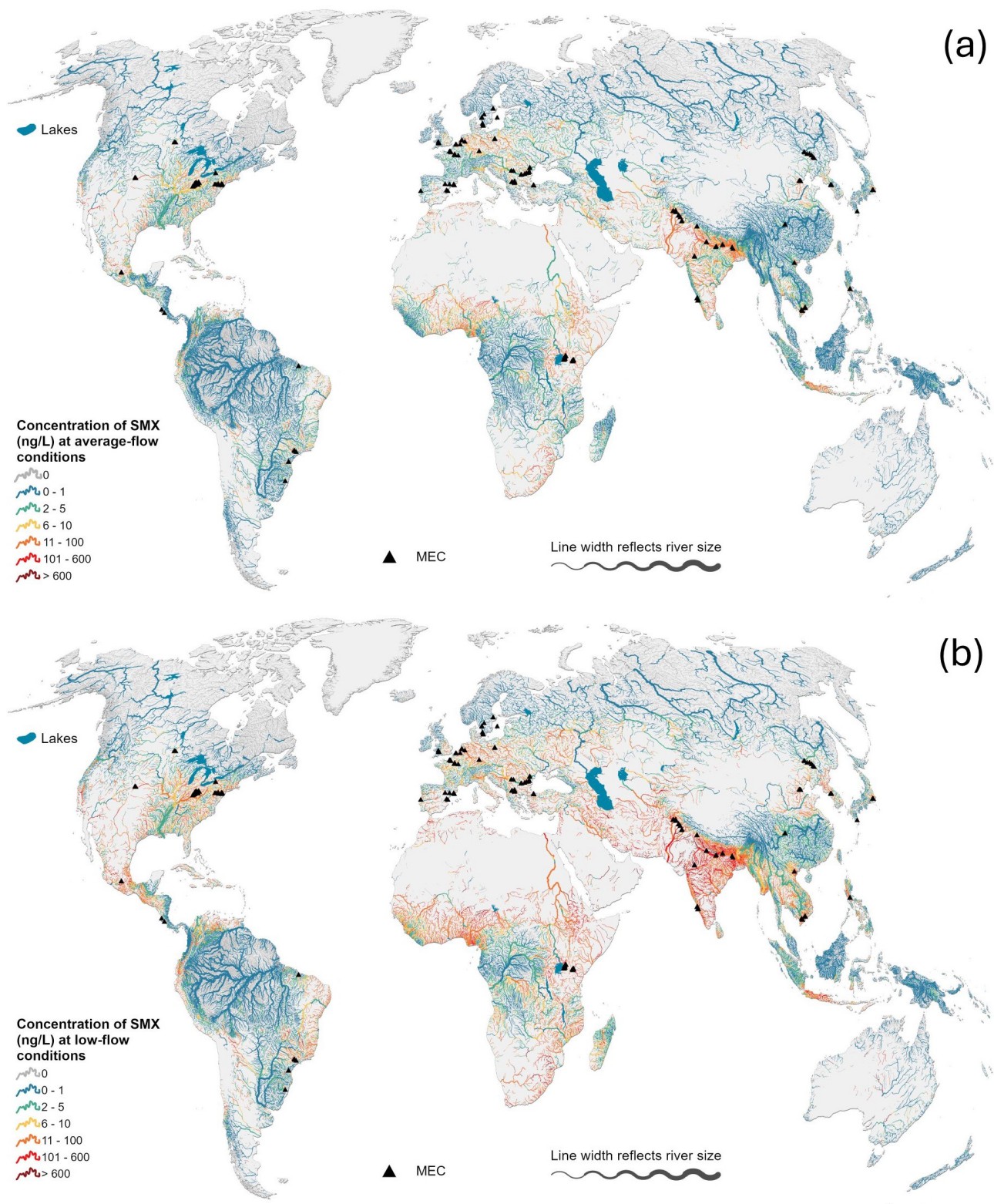

**Figure 3.** Estimated concentrations of sulfamethoxazole (SMX) in the global river network using the HydroFATE model for baseline Scenarios 1 (**a**, average-flow conditions) and 2 (**b**, low-flow conditions). See Table 1 for scenario settings. The black triangles represent the locations of measured environmental concentrations (MECs) used in the model evaluation. For visual clarity, only rivers exceeding a long-term average flow of $3\,\mathrm{m^3\,s^{-1}}$ are shown on the map.<sub>TS9</sub>

**Table 2.** Top 20 countries ranked by their predicted emissions of SMX to rivers and lakes, with their population sources, their national consumption, their emission rates by pathway, and their global totals. For a complete list of all countries, see Table S3 in the Supplement.

| Country | Population source (%) | | | Consumption | | Total emission to rivers and lakes (kg yr$^{-1}$) TS5 | Emission to consumption ratio (%) TS6 | Contaminant pathway into rivers or lakes (%) | | |
|---|---|---|---|---|---|---|---|---|---|---|
| | Treated (WWTPs and DWTSs) | Urban untreated | Rural untreated | Total (kg yr$^{-1}$) | Per capita (µg d$^{-1}$) | | | Treated (WWTPs and DWTSs) | Urban untreated | Rural untreated |
| India | 8.0 | 29.2 | 62.7 | 366 000 | 801 | 32 300 | 8.8 | 9.3 | 53.0 | 37.7 |
| United States | 77.1 | 2.5 | 20.5 | 290 000 | 3070 | 27 000 | 9.3 | 84.4 | 4.2 | 11.4 |
| Pakistan | 18.5 | 23.1 | 58.3 | 223 000 | 3155 | 19 100 | 8.6 | 22.1 | 43.3 | 34.6 |
| South Africa | 47.7 | 28.2 | 24.1 | 102 000 | 6216 | 10 700 | 10.6 | 46.2 | 42.8 | 11.1 |
| Nigeria | 8.3 | 37.3 | 54.3 | 82 300 | 1236 | 7960 | 9.7 | 8.8 | 61.8 | 29.5 |
| Brazil | 61.8 | 17.0 | 21.2 | 77 400 | 1189 | 7910 | 10.2 | 61.7 | 26.7 | 11.6 |
| Indonesia | 5.0 | 52.5 | 42.5 | 74 800 | 904 | 8610 | 11.5 | 4.4 | 73.1 | 22.5 |
| Mexico | 78.4 | 4.3 | 17.4 | 63 000 | 1504 | 6090 | 9.7 | 82.8 | 7.0 | 10.2 |
| Egypt | 65.8 | 25.2 | 8.9 | 55 500 | 1920 | 6220 | 11.2 | 60.0 | 36.1 | 4.0 |
| Ethiopia | 1.2 | 20.9 | 77.9 | 40 400 | 1214 | 2940 | 7.3 | 1.7 | 45.9 | 52.5 |
| DR Congo | 0.3 | 28.4 | 71.4 | 39 900 | 1236 | 3220 | 8.1 | 0.3 | 56.2 | 43.4 |
| Bangladesh | 5.0 | 42.4 | 52.6 | 39 100 | 714 | 4100 | 10.5 | 4.9 | 64.7 | 30.4 |
| Iran | 27.2 | 18.4 | 54.5 | 34 700 | 1265 | 3010 | 8.7 | 31.9 | 33.9 | 34.2 |
| Vietnam | 1.0 | 54.6 | 44.3 | 33 000 | 1031 | 3800 | 11.4 | 0.9 | 76.4 | 22.7 |
| Russia | 79.9 | 0.2 | 19.9 | 32 700 | 696 | 3020 | 9.2 | 88.3 | 0.3 | 11.4 |
| China | 57.3 | 8.0 | 34.7 | 31 800 | 67 | 2860 | 9.0 | 65.0 | 14.2 | 20.8 |
| Ecuador | 60.4 | 1.7 | 37.8 | 22 800 | 4126 | 1970 | 8.7 | 71.1 | 3.2 | 25.7 |
| Myanmar | 1.2 | 30.6 | 68.2 | 22 700 | 1292 | 2070 | 9.1 | 1.3 | 53.7 | 45.0 |
| Germany | 99.1 | 0.0 | 0.9 | 22 300 | 816 | 2270 | 10.1 | 99.6 | 0.0 | 0.4 |
| Tanzania | 0.2 | 28.5 | 71.3 | 21 600 | 1267 | 1700 | 8.0 | 0.2 | 57.0 | 42.8 |
| Total | 34.4 | 21.0 | 44.5 | ~~1 670 000~~ TS7 | 1708 | 160 000 TS8 | 9.4 | 37.6 | 37.9 | 24.5 |
| Global | 36.1 | 19.9 | 44.0 | 2 400 000 | 1331 | 220 000 | 9.2 | 38.2 | 36.0 | 25.7 |

**Table 3.** Top 20 countries by total length of rivers with a predicted risk quotient (RQ) $\geq 1$ for SMX for Scenarios 2 (low-flow conditions) and 1 (average-flow conditions). See Table 1 for scenario settings. The total length of rivers is extracted for each country from the RiverATLAS database (Linke et al., 2019), accounting for all rivers in the world with long-term annual average discharge above 0.1 m$^3$ s$^{-1}$ (i.e., a global total of $23.9 \times 10^6$ km). The increase in length of rivers presenting risk of exposure based on specific conditions was calculated by running the model for the pertinent scenario but changing the parameters and configurations accordingly. See Table S4 in the Supplement for a complete list of all countries.

| Country | Total length | RQ $\geq 1$ at low-flow conditions | | | | RQ $\geq 1$ at average-flow conditions | | | |
|---|---|---|---|---|---|---|---|---|---|
| | of analyzed rivers (km) TS10 | Length of rivers (km) | % of total length | % increase in length without instream decay | % increase in length without lake removal | Length of rivers (km) | % of total length | % increase in length without instream decay | % increase in length without lake removal |
| India | 776 000 | 123 000 | 15.9 | 6.9 | 17.9 | 3370 | 0.4 | 17.4 | 37.1 |
| Pakistan | 102 000 | 34 200 | 33.5 | 6.9 | 4.0 | 8750 | 8.6 | 2.9 | 4.1 |
| Sudan | 100 000 | 15 100 | 15.1 | 7.7 | 1.4 | 290 | 0.3 | 34.5 | 18.6 |
| Iran | 202 000 | 14 500 | 7.2 | 6.3 | 2.3 | 800 | 0.4 | 3.9 | 1.4 |
| Ethiopia | 186 000 | 14 400 | 7.7 | 10.7 | 2.6 | 382 | 0.2 | 8.9 | 21.7 |
| South Africa | 107 000 | 13 900 | 13.0 | 10.8 | 43.0 | 3770 | 3.5 | 27.6 | 68.2 |
| Nigeria | 201 000 | 12 600 | 6.3 | 7.2 | 11.3 | 673 | 0.3 | 48.6 | 85.9 |
| Saudi Arabia | 71 400 | 11 800 | 16.5 | 12.0 | 0.3 | 67 | 0.1 | 35.8 | 0.0 |
| United States | 1 780 000 | 10 700 | 0.6 | 13.5 | 30.9 | 1240 | 0.1 | 17.7 | 69.1 |
| Mexico | 270 000 | 9180 | 3.4 | 9.8 | 30.0 | 1180 | 0.4 | 7.7 | 19.3 |
| Algeria | 94 200 | 8760 | 9.3 | 5.8 | 9.9 | 806 | 0.9 | 16.4 | 15.0 |
| Yemen | 23 100 | 8540 | 37.0 | 2.0 | 0.1 | 1210 | 5.3 | 5.7 | 1.3 |
| Niger | 49 500 | 7990 | 16.1 | 8.7 | 7.9 | 444 | 0.9 | 25.2 | 73.2 |
| Somalia | 42 200 | 7440 | 17.6 | 8.3 | 0.0 | 32 | 0.1 | 34.4 | 0.0 |
| Iraq | 36 400 | 6100 | 16.8 | 3.9 | 7.4 | 171 | 0.5 | 19.9 | 1.8 |
| China | 1 440 000 | 5900 | 0.4 | 15.8 | 20.7 | 39 | 0.0 | 38.5 | 176.9 |
| Chad | 85 100 | 5870 | 6.9 | 12.5 | 2.4 | 68 | 0.1 | 0.0 | 79.4 |
| Afghanistan | 79 700 | 4640 | 5.8 | 9.9 | 2.2 | 56 | 0.1 | 0.0 | 16.1 |
| Oman | 14 000 | 4620 | 33.0 | 3.2 | 0.1 | 0 | 0.0 | 0.0 | 0.0 |
| Turkmenistan | 14 900 | 3920 | 26.3 | 4.0 | 11.9 | 402 | 2.7 | 8.2 | 2.2 |
| Total | ~~5 680 000~~ TS11 | ~~323 000~~ TS12 | 5.7 | 7.8 | 12.9 | ~~23 700~~ TS13 | 0.4 | 13.1 | 27.8 |
| Global | 23 900 000 | 409 000 | 1.7 | 8.4 | 14.4 | 29 000 | 0.1 | 12.8 | 26.8 |

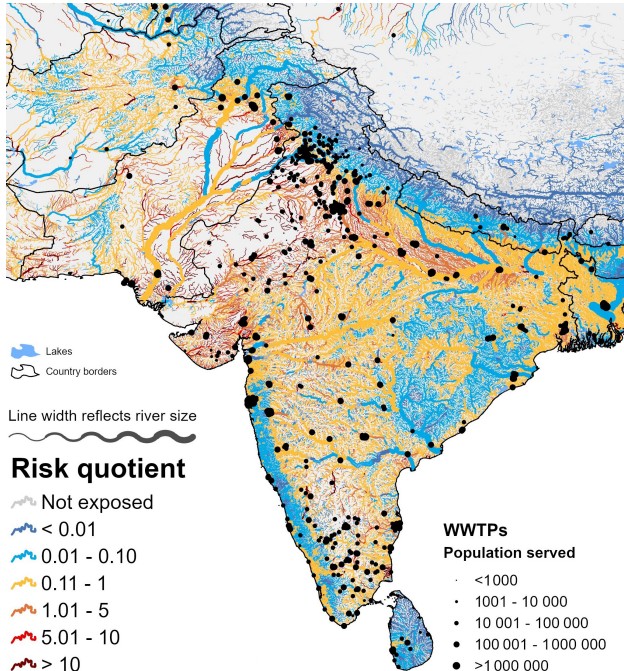

**Figure 4.** Estimated risk quotient for sulfamethoxazole (SMX) using HydroFATE in parts of Southeast Asia, with a focus on India and Pakistan. The risk quotient was calculated using Scenario 2 (see Table 1 for scenario settings). The black dots represent the location of wastewater treatment plants (WWTPs) in the database HydroWASTE. For visual clarity, only rivers exceeding a long-term average flow of $0.1\,\mathrm{m^3\,s^{-1}}$ are shown on the map.

if both river and lake environmental decay processes were omitted, there would be a combined increase of 22.8 % and 39.6 % in the length of rivers that fall in the risk category compared to Scenarios 2 and 1, respectively (Table 3). If only lake removal processes were excluded, there would be an increase of 14.4 % in Scenario 2 and 26.8 % in Scenario 1. Lake removal is predicted to be particularly important in rivers in South Africa, United States, Mexico, and China. For instance, without lake removal, there would be an increase of 43 % of rivers in South Africa falling in the risk category at low-flow conditions.

Finally, to demonstrate the utility of a contaminant fate model operating at high spatial resolution, Fig. 4 depicts the risk distribution under low-flow conditions (Scenario 2) for the region of Southeast Asia, including the two countries (India and Pakistan) with the longest total length of rivers in which SMX concentrations exceed PNEC. The high spatial resolution permits the detection of local increases in risk immediately downstream of individual WWTPs, which then can diminish along the flow paths once inflowing tributaries cause dilution effects. Model results also reveal the exposure of individual rivers receiving contaminant discharge without any treatment (i.e., areas without any black dots but presenting a high density of rivers at risk).

## 4.4 Performance evaluation

Modelling results were evaluated by comparing predicted SMX concentrations with available measurements in river reaches across the world using 227 MECs with values above the detection threshold and 134 measurements below the limits of detection. Figure 5a to d show CE1 an analysis of results for baseline Scenario 1 (average-flow conditions; see Table 1 for scenario settings), distinguished by certain characteristics of the measurements. The different colors of points in the overall scatter plot shown in Fig. 5a illustrate the global distribution of measurements. The African continent presents the highest SMX concentrations (both measured and predicted) and the predicted concentrations in Asia, Europe, and Central America are in their majority below reported measured concentrations. These results confirm that model predictions for Scenario 1 are generally reasonable, with 77.5 % of the predicted values being within 1 order of magnitude of the measured concentrations reported in literature (Johnson et al., 2008; Oldenkamp et al., 2018).

Figure 5b shows substantial uncertainties in PEC calculations (i.e., reflected by the extent of error bars) when using the parameter and configuration settings of Scenarios 3 and 4, representing low-end and high-end simulations that were within plausible ranges. Taking these uncertainties into account, 80 % of all MECs fell within the range of error bars of HydroFATE; that is, they were reproducible by the model within at least one of the chosen parameter and configuration settings. Additional model performance indicators, obtained by comparing all 227 MECs against estimates derived for Scenario 1, also reflect overall fair results yet with a clear bias of modelled concentrations tending to be lower than reported measurements.

Figure 5c shows the same points with bubbles sized according to average discharge at the measurement location, and Fig. 5d shows bubbles sized according to upstream urban extents. Most measurement locations for which model predictions of concentrations were too high are located downstream of urban populations on rivers with low discharge, which is a challenging combination to model; that is, urban streams can be heavily modified by anthropogenic activities that influence their flow quantities and water quality, such as channelization, dams, and sewers. Besides, if the urban population is not served by WWTPs, the predictions were based on the assumption of a constant direct discharge coefficient, which in all probability is variable in reality. Despite these uncertainties, only two measurements were predicted erroneously above the PNEC threshold, resulting in a risk quotient that is falsely predicted to be above 1. On the other hand, 17 measurements were erroneously predicted to be below the PNEC threshold, which is in accordance with the overall conservative approach and scenario configuration used in this case study.

In addition to the performance evaluation presented in Fig. 5, 63 % of PECs at the same locations as MECs that re-

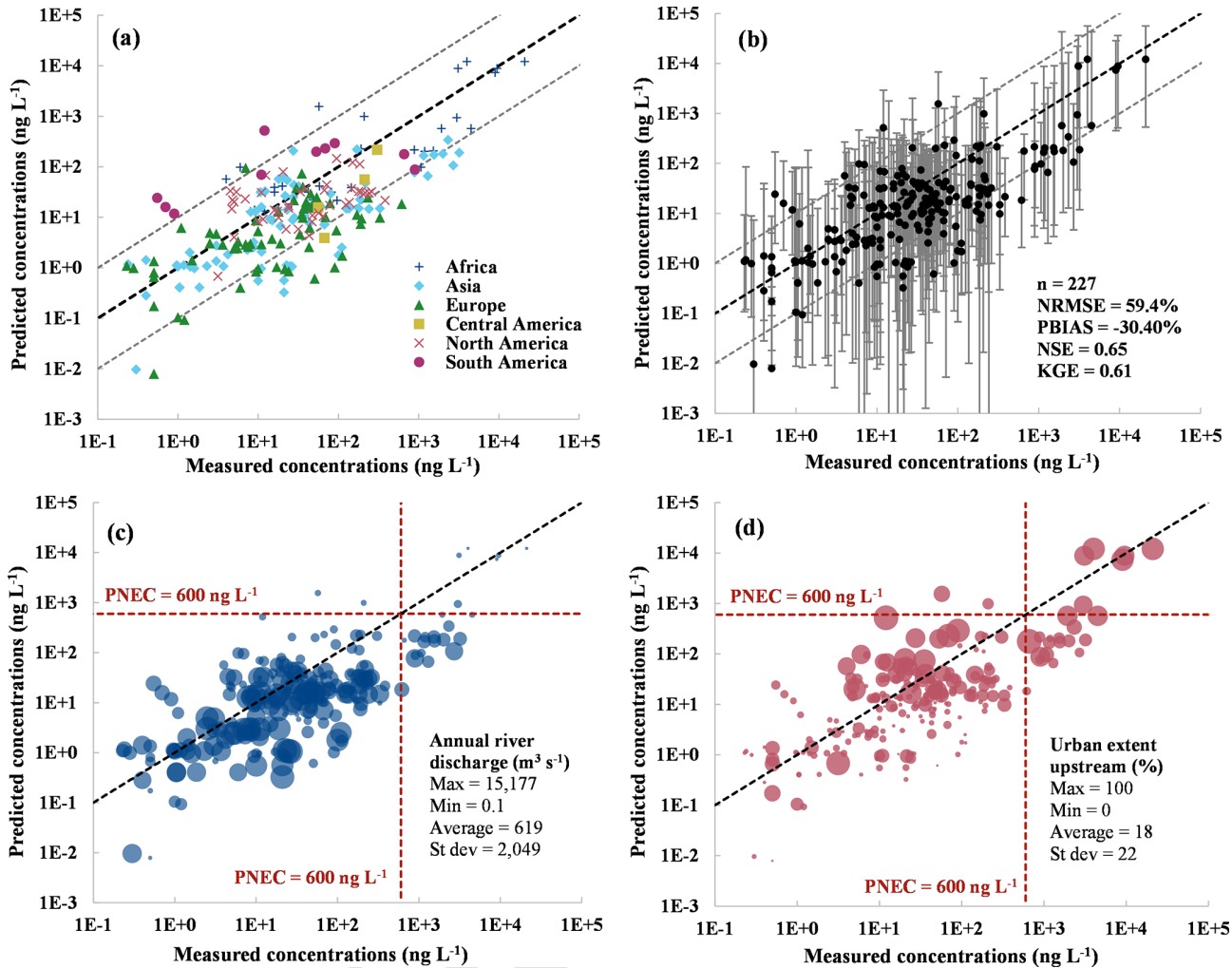

**Figure 5.** Evaluation of the SMX concentrations estimated using HydroFATE for baseline Scenario 1 (average-flow conditions; see Table 1 for scenario settings). Individual points represent comparisons between measured (MEC) and predicted (PEC) environmental concentrations in the same river reach. The black dashed line represents the 1 : 1 line and the gray dashed lines represent the error line corresponding to 1 order of magnitude. Panel **(a)** shows a general scatter plot between MECs and PECs and indicates their spatial distribution (with different symbols and colors representing the geographic regions). Panel **(b)** shows uncertainties of PECs where the error bars represent the range of resulting PECs using the parameter and configuration settings of Scenarios 3 and 4; *n* is the number of records; NRMSE is the normalized root mean square error; PBIAS is the percent bias; NSE is the Nash–Sutcliffe efficiency; KGE is the Kling–Gupta efficiency. Panel **(c)** indicates the annual river discharge at the river reach of the MEC in different bubble sizes; panel **(d)** indicates the urban extent upstream of the river reach of the MEC in different bubble sizes; the red dashed lines represent the threshold for the PNEC. Error bars in panel **(b)** that extend below 0.001 ng L$^{-1}$ may include predicted zero concentrations.

ported to be below detection limits were correctly predicted to have concentrations that fell below the reported limits. If allowing an error of 1 order of magnitude, the success rate increased to 91 %. Three additional scenarios (including 14 sub-scenarios) were also analyzed to specifically test the uncertainty ranges introduced by individual parameter and configuration settings (see Sect. S2 in the Supplement for details). The results indicate that the model reacts sensitive yet within reasonable boundaries to the permutations of individual parameter settings. Of particular importance are the settings related to substance removal simulations in the model

which depend on local characteristics and on the dominating pathway of the contaminant.

## 5   Discussion

### 5.1   HydroFATE: strengths and limitations

In this study, the first global application of the contaminant fate model (CFM) HydroFATE was presented, building upon previous stages of the model that were used to assess the distribution of pharmaceuticals at the regional scale in Canada,

China and India. One of the main characteristics that distinguishes HydroFATE from other global CFMs, besides its high spatial resolution, is that contaminant pathways can be differentiated based on whether the wastewater undergoes treatment or not and, if so, at what treatment level. Contaminants generated by populations connected to wastewater facilities are partially removed by treatment processes, whereas contaminants generated by populations not connected to any wastewater system are assumed to undergo natural attenuation processes, which in the case of rural populations also depend on how distant they are from any waterbody. The model application showed that different regions indeed responded differently to pharmaceutical drug consumption depending on the main pathway of the contaminant before reaching the river system.

Despite its high spatial resolution, HydroFATE has primarily been designed as a CFM that operates at large scales, including at the global scale, and it can be readily applied with existing input data. Due to the necessary model simplifications to enable such an approach, it is recognized that, even with anticipated future refinements, substantial uncertainties will remain with respect to the model's predictive capability. As such, HydroFATE is intended to serve as a screening model whose primary purpose is to identify critical areas where detailed field studies should be performed.

To apply any model appropriately and interpret its output, it is essential to understand its limitations. The main limitations of HydroFATE stem from its steady-state approach; the difficulty of capturing some underlying processes at the global scale; the lack of information on the behaviour, use, and disposal of most CECs; and unaccounted variability regarding most input parameters of the model. HydroFATE transport processes are based on long-term average discharge or long-term monthly minimum discharge, which does not account for the seasonality of river flows or any shorter-term fluctuations that affect the dilution capabilities (or lack thereof) of contaminant concentrations. The decay of contaminants along rivers and in lakes is assumed to follow a first-order process, lumping and simplifying complex processes such as deposition, adsorption, photodegradation, and bioaccumulation that occur over time. These processes also depend on local environmental and biological characteristics that are currently very difficult to capture on a global scale. Therefore, more experiments and measurements are needed to reduce the uncertainties inherent in quantifying the decay constants for different substances and under different conditions, especially contaminants of emerging concern. In the presented case study application, an average decay constant arising from only a few reports for sulfamethoxazole was used, which likely represents an overly narrow range that does not adequately capture what happens under different conditions in rivers worldwide.

The efficiency of a WWTP in removing a specific contaminant is also a complex process that depends on characteristics of the individual facilities and local conditions that are not represented in the global HydroWASTE database. Furthermore, processes not simulated by HydroFATE may have an impact on contaminant loads entering surface waters. For example, depending on how far a household is located from the facility, decay processes in sewers can reduce contaminant loads on the way to the WWTP. In addition, sewer lines that are poorly maintained may result in wastewater leakages into the ground, further reducing the load of contaminant before it reaches the WWTP.

On the other hand, wastewaters from almost half of the world's population are untreated. The uncertainties in HydroFATE related to contaminant simulations from untreated sources have two main sources: (1) the difficulty to spatially distinguish wastewater contributions from populations as treated or untreated in the first place, based only on WWTP characteristics, country-level statistics, and a global population grid, and (2) the generalization of the pathways of untreated contaminants into only two types (i.e., distinguishing only rural versus urban conditions), based on simplified assumptions and very little evidence from field experiments (Grill et al., 2018). In fact, Grill et al. (2018) found in a sensitivity analysis for China that the setting of the direct discharge coefficient in rural areas represented the main source of model uncertainties. However, while the simplified approach to modelling soil-related processes and the corresponding determination of spatially heterogenous parameter settings are major limitations of the HydroFATE model and likely important sources of error, in particular in areas dominated by untreated pathways, these simulations are critically important to be implemented in the model design. For example, in the presented case study, the untreated pathways contributed an estimated 62 % of the global emission of sulfamethoxazole, demonstrating their decisive role. Overall, despite the described uncertainties related to simplified process simulations, the baseline scenario was able to reproduce field measurements reasonably well, especially considering the large range of possible values for the direct discharge coefficients (see Table 1). Panel (d) of Fig. 5 suggests a general overestimation of contaminant concentrations in regions with substantial urban extents upstream (larger bubbles) and a general underestimation for rural areas (i.e., smaller bubbles representing areas with smaller urban extents), an observation which could be used to revise the direct discharge coefficients in future model runs. However, to ensure that HydroFATE is generally applicable to a range of substances, it is recommended that the model be first tested when applied to other substances before a potential calibration of different direct discharge coefficients is carried out to improve model performance. In addition, as further discussed below, the current model version only accounts for one type of source (domestic), which excludes veterinary and industrial contributions that can be present in waters, making any uncontrolled measurements inadequate for calibration.

Besides the river network, the pathways of the contaminants determine most of the spatial contaminant distribution

of the model. The method developed in this study distinguishes the wastewater contributions from the global population as treated or untreated by relying mostly on global population and urban extent grids, as well as the global WWTP database HydroWASTE. Both population and urban extent grids have their own uncertainties related to the way they have been developed, their spatial resolutions, and their representative years, potentially misrepresenting actual conditions especially in sprawling cities in developing countries (Sridhar and Mavrotas, 2021). HydroWASTE is a data compilation that contains estimated characteristics instead of official records for 9 % of the WWTPs, and it does not include small DWTSs that are more common in rural areas. In the absence of data, country-level statistics on sanitation were used to minimize these uncertainties regarding HydroWASTE.

## 5.2 Performance evaluation of HydroFATE using sulfamethoxazole

As a first case study application of HydroFATE, country-level consumption data of sulfamethoxazole (SMX) were used to assess its distribution in the global river and lake network. Results predicted that a total of approximately 214 000 kg of SMX is released into rivers and lakes every year from domestic sources. A cursory exposure assessment shows that this release may potentially result in a risk of environmental impact (i.e., defined as PEC $\geq$ PNEC) during low-flow conditions throughout 390 000 km of the global river network.

In terms of the input information regarding SMX, uncertainties can derive from various assumptions incorporated into the model parameters, including the country-level consumption rate, the excretion fraction (after metabolism), the wastewater treatment removal efficiency, and the instream decay constant. The contaminant emission is estimated based on country averages of consumption and population density based on the assumption that every person consumes the same amount of SMX in a year across a country, which therefore does not account for regional, municipal, or personal (e.g., age-dependent) spatial variability of consumption. The excretion fraction has a relatively small range of uncertainty (Table 1) as the metabolism process of SMX inside the human body is well known and was extensively studied by pharmaceutical companies before its release during the drug development phase (Zhang and Tang, 2018). The removal efficiency in treatment facilities, including WWTPs and DWTSs, depends on the specific type of treatment being employed. The values reported in the literature vary widely, possibly due to SMX being transformed to N4-acetyl-SMX and glucuronide conjugates (the most common SMX metabolites) and vice versa during the treatment process (Straub, 2016). HydroFATE is, in principle, able to account for different levels of treatment provided by WWTPs (i.e., primary, secondary, or advanced) by using different removal efficiencies. However, due to a lack of consistent

data, a choice was made in the presented case study to apply one single average value for the substance removal efficiency across all wastewater facilities, including DWTSs. This could lead to over- or underestimated SMX concentrations in rivers, since primary and advanced treatment processes are expected to result in lower or higher removal efficiencies, respectively.

To evaluate the general performance of HydroFATE regarding its simulation of SMX concentrations, PECs resulting from the model were compared with MECs reported in the literature. The results showed an overall reasonable predictive capability with the goodness-of-fit indicators NSE and KGE above 0.6 and with 77.5 % of PECs being within 1 order of magnitude of reported MECs. This was despite the inherent uncertainties associated with assumptions made in the development of the model and those associated with estimates of the various model parameters and input datasets. It is noted that other global water quality models, which also simulate substance loads and concentrations, have reported similar values of NSE between 0.4 and 0.71 (Font et al., 2019; Harrison et al., 2019). However, a more detailed comparison between results from these models and HydroFATE is difficult as different substances and spatial resolutions were applied.

Unfortunately, the lack of specificity of field measurements, for which literature sources generally do not provide enough information on the precise locations of measurements nor river discharge conditions at the time of sample collection, does not allow for a conclusive evaluation of the model under different modelling scenarios. Since there is a possibility that some of the MECs were measured during low-flow conditions (Sect. 4.1.3), the comparison with results using Scenario 1 (i.e., representing average-flow conditions) might not be appropriate. This issue is further explored in Fig. 6, where error bars were added to the points of the scatter plot between MECs and PECs for Scenario 1. The ends of the error bars (extending only leftwards) show a recalculated MEC for average discharge conditions when assuming that the original concentration of SMX was measured during low-flow conditions. This analysis demonstrates that cases in which predicted values were too low could, in part, be explained by uncertainties within the measurements rather than errors in the model predictions.

Importantly, the PECs simulated by the present version of HydroFATE are limited in that they do not include contaminant contributions from veterinary use or pharmaceutical manufacturing operations due to a lack of available data. As it is not possible to isolate only the contribution from domestic sources in the MECs, this uncertainty in MECs combined with the omission of veterinary antibiotics in the simulated PECs could explain a portion of the high negative bias found in the evaluation. Once data on veterinary use or manufacturing become available, they could readily be implemented to refine HydroFATE.

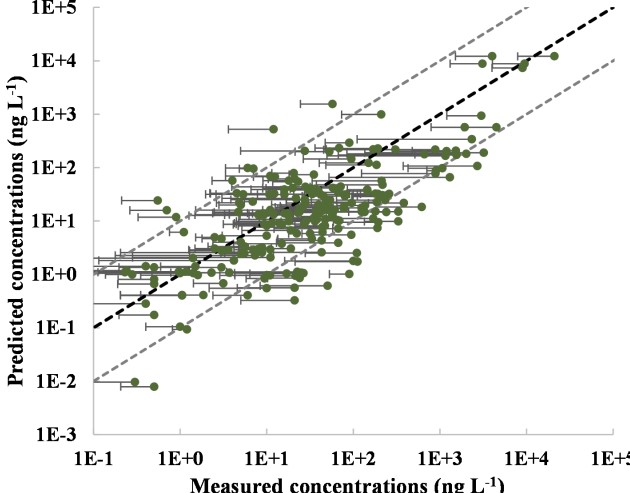

**Figure 6.** Estimated uncertainties of measured environmental concentrations (MECs) considering the lack of reported river discharge conditions at the time of collection. Individual points represent comparisons between MECs and predicted environmental concentrations (PECs) estimated using HydroFATE for baseline Scenario 1 in the same river reach. The black dashed line represents the 1 : 1 line, and the gray dashed lines represent the error line corresponding to 1 order of magnitude. The ends of the error bars represent a recalculated MEC for average discharge conditions assuming the original was measured during low-flow conditions.

## 6 Conclusion

Despite its current shortcomings and inherent uncertainties, HydroFATE is the most spatially detailed global CFM currently available. It tracks multiple pathways of contaminants in the river and lake environment and has the potential to be used for any CEC of domestic use. In its current version, HydroFATE is expected to be particularly useful to identify specific areas in the river network where high concentrations of contaminants may be found. As such, potential applications include the support of decision-making in order to prioritize and focus resources, regarding: (1) locations that should be the subject of detailed field measurements and local environmental impact studies; (2) the creation of scenarios for policy-making and management of water resources at regional or international scales; (3) the development of screening methods to inform new regulation or guidelines for the pharmaceutical industry with respect to establishing markets for their products and performing regulatory compliance tests to safeguard ecosystems and human health; (4) the development of new or updated treatment standards for contaminants of emerging concern, including the establishment of design specifications for wastewater treatment systems in specific regions; and (5) the deployment of new treatment technologies.

## Appendix A: WWTP service areas

### A1 Delineation of WWTP service areas

Figure A1 shows the conceptual design of the method developed to delineate wastewater treatment plant (WWTP) service areas for every WWTP of the HydroWASTE database (Ehalt Macedo et al., 2022), using a population grid (World-Pop; WorldPop and CIESIN, 2018) combined with an urban versus rural classification (Pesaresi and Freire, 2016) (see Sect. 2.3 TS14 for more details on data sources). In the first of a total of six iterative processing steps, every population pixel located within 10 km of any WWTP was temporarily assigned to the closest WWTP by creating Thiessen polygons around all WWTP point locations, where a Thiessen polygon defines the area that is closer to its associated point than to any other point. Then, a rank value was calculated for every population pixel inside each Thiessen polygon indicating its assumed likelihood to be associated with the respective WWTP (see Box A1 for calculations). The ranking assumed that WWTPs tend to serve populations in the following order of priority (from highest to lowest): (1) residents in closer vicinity to the WWTP, (2) residents in areas of high population density, (3) residents of urban areas (versus rural areas), and (4) residents living in clustered/contiguous areas (versus dispersed single pixels).

After ranking all pixels within each Thiessen polygon, they were gradually assigned to their respective WWTP until the summed population was equivalent to the value of "population served" reported in the WWTP database. After completion of this population assignment, dispersed single pixels or minor clusters were removed if they were not part of the largest contiguous area and did not form their own additional area of at least 9 pixels, assuming that small, isolated population centres are not prioritized to be connected to a WWTP. If a WWTP's "population served" was reached at the end of this first iteration, the WWTP was assumed to be "filled up" and its assigned population pixels were removed from the population map. All remaining pixels were classified to be unassigned.

Next, four additional iterations were performed aiming to fill up the remaining WWTPs. In each of these iterations, every unassigned population pixel was temporarily assigned to the closest WWTP that was not yet filled up; that is, the pixels were temporarily assigned by creating new Thiessen polygons around the remaining WWTPs by using increasingly larger distance thresholds of 20, 30, 40, and 50 km, respectively. The same ranking system (Box A1) was used to permanently assign pixels to the remaining WWTPs. However, an additional constraint was applied in each of the four iterations to avoid excessive service area distances for smaller WWTPs: that is, WWTPs serving less than 10 000 people were not considered in the second iteration, even if they were not yet filled up; WWTPs serving less than 100 000 people were not considered in the third iteration; WWTPs serving

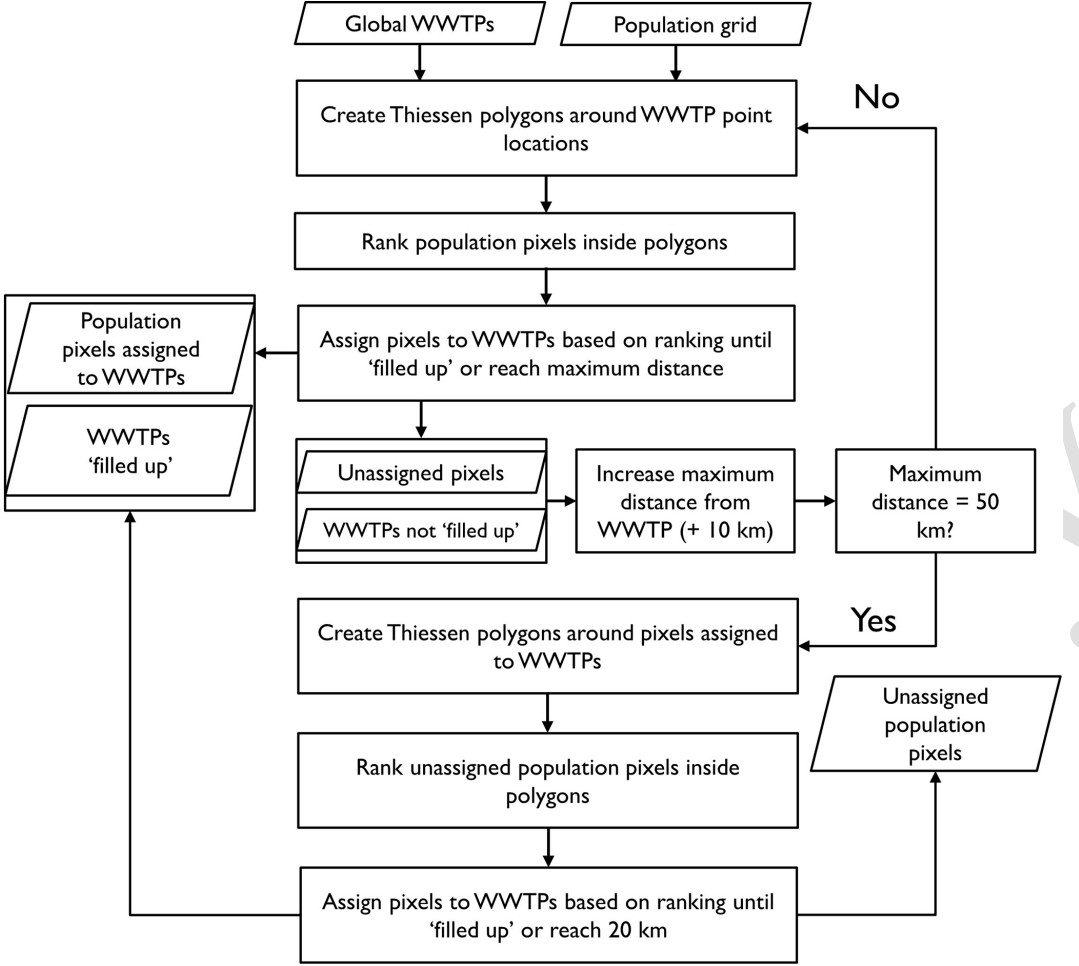

**Figure A1.** Conceptual approach of delineating service areas for wastewater treatment plants (WWTPs).

less than 600 000 people were not considered in the fourth iteration; and WWTPs serving less than 1.1 million people were not considered in the fifth iteration.

After these five iterations (corresponding to a maximum distance of 50 km), one final iteration was performed for all WWTPs that were still not filled up (even the smaller WWTPs serving less than 10 000 people). That is, all remaining unassigned pixels within their Thiessen polygon and up to 20 km from the WWTP's current service area (i.e., from the result of the previous iterations) were ranked and assigned to the respective WWTP, even if they were not contiguous to other pixels already assigned. This additional iteration ensures that remaining unassigned pixels in the proximity of WWTPs of any size not yet filled up have a final opportunity to be assigned, including those pixels that were closer to other WWTPs in earlier iterations but were ultimately not assigned to them.

## A2   Evaluation of resulting WWTP service areas

The population served by WWTPs as spatially assigned by the procedure developed here is by design equal to or lower than the population served as reported in the HydroWASTE database, which is confirmed in Fig. A2. That is, the described procedure delivers the best estimate yet with an intended bias towards underestimating the amount of people served by WWTPs. This design was intentionally chosen to avoid exceeding reported values of populations served while allowing for underestimates which may represent various plausible realities, such as cases in which reported population numbers represent maximum WWTP capacities. From the total 45 348 original points of WWTP locations used in this study, 44 495 (98 %) had their population served assigned within 1 order of magnitude from reported values, with an $R^2$ (coefficient of determination) of 0.96 and a bias (percent error) of $-13.6$ %. Figure A2 shows that the largest discrepancies were found for smaller WWTPs that are reported to serve less than 10 000 people, likely including cases where WWTPs treats industrial wastewaters or serves areas

A total rank value ($rank_{T,m}$) is calculated for every pixel $m$ inside the Thiessen polygon associated with each WWTP based on three criteria: the distance of the pixel to the WWTP ($rank_{D,m}$), the pixel's population count ($rank_{P,m}$), and whether the pixel is located in an urban or rural area ($rank_{urb,m}$), following equations A1 to A3:

The value of $rank_{D,m}$ (dimensionless) is normalized between 0 and 100, using the equation:

$$rank_{D,m} = D_{WWTP,m}^{-0.8} \qquad (A1)$$

where $D_{WWTP,m}$ is the distance between pixel $m$ and the WWTP in decimal degrees.

The value of $rank_{P,m}$ (dimensionless) is also normalized between 0 and 100, using the equation:

$$rank_{P,m} = 20 \times \log_{10} P_m \qquad (A2)$$

where $P_m$ (persons) is the number of people in pixel $m$. Note that only pixels with a population count larger than 10 are assigned to WWTPs.

The value of $rank_{urb,m}$ (dimensionless) is assigned to be 0 for rural areas and 100 for urban areas, according to the urban extent grid.

Finally, the total rank ($rank_{T,m}$; dimensionless) is calculated for each pixel $m$ as:

$$rank_{T,m} = (0.5 \times rank_{D,m}) + (0.25 \times rank_{P,m}) + (0.25 \times rank_{urb,m}) \qquad (A3)$$

**Box A1.** Ranking method to prioritize the likelihood of a population pixel to be associated with a WWTP. The ranking is established for all population pixels inside the Thiessen polygon that surrounds the WWTP. TS15

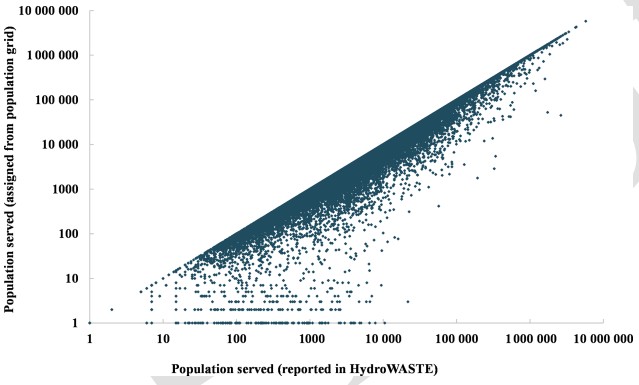

**Figure A2.** Evaluation of the method used to spatially allocate populations from a global population grid to the WWTPs of the HydroWASTE database.

**Table A1.** Averages of estimated service area extents by WWTP size as reported in the HydroWASTE database (in terms of population served).

| Population served (HydroWASTE) | Average service area extent (km$^2$) |
|---|---|
| 1–100 | 0.2 |
| 101–1000 | 1.6 |
| 1001–10 000 | 20.7 |
| 10 001–100 000 | 42.4 |
| 100 001–1 000 000 | 133.4 |
| > 1 000 000 | 406.6 |

with substantial transient population (e.g., tourists, workers), which are not represented in the population grid. Only two WWTPs with reported capacities of more than 1 million people showed an underestimation due to our service area allocation of more than 1 order of magnitude. Both are located near a village in Poland with less than 2000 residents and are likely the result of reporting errors.

Table A1 shows the averages of the service area extents (in km$^2$) resulting from the described allocation method for different reported sizes of WWTPs. For comparison, the WWTP of Montréal, the largest in North America, serves most of the population on the island of Montreal ($\sim 2$ million people) which covers an area of 473 km$^2$ (source: city of Montréal, Quebec CE2, Canada).

*Code and data availability.* Model predictions for the four main scenarios were obtained with a run time of 18 min using a desktop PC with Intel Core i7-10700 CPU, 2.90 GHz, and 32 GB of RAM. A license for the software ArcGIS Pro (by Esri) is required to run the provided scripts. The code for HydroFATE v1 (including compiling instructions) is available under the GNU General Public License v3.0, and the input and output data are available under a CC-BY-4.0 License at the following URL: https://doi.org/10.6084/m9.figshare.23646282 (Ehalt Macedo et al., 2023). Under the same URL, the code for the delineation of WWTP service areas and the resulting grid containing all contaminant pathways are provided. Finally, the global river network dataset and the

associated attribute information for every river reach as well as the results from the case study are also available at the same URL under a CC-BY-4.0 License.

*Supplement.* The supplement related to this article is available online at: https://doi.org/10.5194/gmd-17-1-2024-supplement.

*Author contributions.* HEM: conceptualization, methodology, software, validation, formal analysis, investigation, data curation, writing (original draft), visualization. BL: conceptualization, methodology, resources, writing (review and editing), supervision, funding acquisition. JN: conceptualization, methodology, resources, writing (review and editing), supervision, funding acquisition. GG: conceptualization and methodology of original model, software.

*Competing interests.* The contact author has declared that none of the authors has any competing interests.

*Acknowledgements.* We thank Eili Y. Klein from the Center for Disease Dynamics, Economics & Policy for providing the country-level averages of annual consumption per capita of SMX. We also thank Ranish Shakya for the contribution on early model development stages of HydroFATE.

*Financial support.* This research has been supported by the Natural Sciences and Engineering Research Council of Canada (NSERC Discovery grant nos. RGPIN/04541-2019 and RGPIN/03792-2016) and the James McGill Chair program of McGill University. TS18

*Review statement.* This paper was edited by Lele Shu and reviewed by Francesco Bregoli and three anonymous referees.

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

**Remarks from the language copy-editor**

CE1   "show" as a plural verb is correct here according to our standards. The verb agrees with the figure panels (labelled 5a to 5d). Please confirm.

CE2   Note that "city" should not be capitalized; "City of Montréal" is not considered a proper noun. Do you mean "Montreal City Council" or something similar? We use "Quebec" without accents of the province name (and with accents for the city name).

**Remarks from the typesetter**

TS1   Please note: the Supplement has not been exchanged since there was no comment about it, and the file you provided this proofreading round seems to be the same as the current Supplement.

TS2   Please confirm the figure.

TS3   Please note: variables that consist of two or more letters are roman according to our standards.

TS4   Thank you for your feedback. Unfortunately, the space cannot be avoided with the center line in columns 4 and 5. Thank you for your understanding.

TS5   Thank you for your feedback. Please note that your suggested change would increase the width of this table, which would require to scale it down further and decrease readability. I suggest to keep the line breaks.

TS6   Please see my previous note.

TS7   Please give an explanation of why this needs to be changed. We have to ask the handling editor for approval. Thanks.

TS8   Please give an explanation of why this needs to be changed. We have to ask the handling editor for approval. Thanks.

TS9   Please confirm the figure.

TS10   Please see my note about the line break in Table 2.

TS11   Please give an explanation of why this needs to be changed. We have to ask the handling editor for approval. Thanks.

TS12   Please give an explanation of why this needs to be changed. We have to ask the handling editor for approval. Thanks.

TS13   Please give an explanation of why this needs to be changed. We have to ask the handling editor for approval. Thanks.

TS14   Please confirm.

TS15   Please confirm the box.

TS16   Please give an explanation of why this needs to be changed. We have to ask the handling editor for approval. Thanks.

TS17   Please give an explanation of why this needs to be changed. We have to ask the handling editor for approval. Thanks.

TS18   Please confirm both Acknowledgements and Financial support sections.