# Peer review of "HydroFATE (v1): A high-resolution contaminant fate model for the global river system"

_EGUsphere, 2023_

## Referee Comment (RC3)

**HydroFATE (v1): A high-resolution contaminant fate model for the global river system**

Heloisa Ehalt Macedo[1], Bernhard Lehner[1], Jim Nicell[2], Günther Grill[1]

5  [1]Department of Geography, McGill University, Montreal, QC H3A 0B9, Canada
[2]Department of Civil Engineering, McGill University, Montreal, QC H3A 0C3, Canada

*Correspondence to*: Heloisa Ehalt Macedo (heloisa.ehaltmacedo@mail.mcgill.ca) and Bernhard Lehner (bernhard.lehner@mcgill.ca)

10  **Abstract.** Pharmaceuticals and household chemicals are neither fully consumed nor fully metabolized when routinely used by humans, thereby resulting in the emission of residues down household drains and into wastewater collection systems. Since treatment systems cannot entirely remove these substances from wastewaters, the contaminants from many households connected to sewer systems are continually released into surface waters. Furthermore, diffuse contributions of wastewaters from populations that are not connected to treatment systems can directly (i.e., through surface runoff) or indirectly (i.e., through soils and groundwater) contribute to contaminant concentrations in rivers and lakes. The unplanned and unmonitored release of such contaminants can pose important risks to aquatic ecosystems and ultimately human health. In this work, the contaminant fate model HydroFATE is presented which is designed to estimate the surface-water concentrations of domestically used substances for virtually any river in the world. The emission of compounds is calculated based on per capita consumption rates and population density.
[Figure]
 global database of wastewater treatment plants is used to separate the effluent pathways from populations into treated and untreated, and to incorporate the contaminant pathways into the river network. The transport in the river system is simulated while accounting for processes of environmental decay in streams and in lakes. To serve as a preliminary performance evaluation and proof of concept of the model, the antibiotic sulfamethoxazole (SMX) was chosen, due to its widespread use and the availability of input and validation data. The comparison of modelled concentrations against a compilation of reported SMX measurements in surface waters revealed reasonable results despite inherent model uncertainties. A total of 390,000 km of rivers were predicted to have SMX concentrations that exceed environmental risk thresholds. Given the high spatial resolution of predictions, HydroFATE is particularly useful as a screening tool to identify areas of potentially elevated contaminant exposure and to guide where local monitoring and mitigation strategies should be prioritized.

**Number: 1**     Author: Anonym     Subject: Highlight    Date: 28/08/2023 18:46:40

Did you do it or is it another project result? Please cite accordingly

**Number: 2**     Author: Anonym     Subject: Highlight    Date: 28/08/2023 18:47:38

which processes? Describe them

**Number: 3**     Author: Anonym     Subject: Highlight    Date: 28/08/2023 18:48:34

SMX is also widely used in vet medicine. Therefore, the use is not only human.

**Number: 4**     Author: Anonym     Subject: Highlight    Date: 28/08/2023 18:49:05

Global model already exists and published here in GMD. But, I think that the resolution and additional processes are indeed added values here.

[revised manuscript text omitted]

Number: 1          Author: Anonym      Subject: Highlight    Date: 24/08/2023 14:38:33

The mechanism of water/soil partitioning (or absorption to soil) and on-soil decay is well understood. The problem is that the variability of the environmental condition is too complex for such global model. In principle you need to know the type of soil, the permeability, the hydrology (for runoff events, leaching and groundwater flow) for each cell of your domain. I understand that this is difficult for such scale. But this needs to be discussed not only on the bases that this is poorly understood.

The parameters that would represent all these processes are here set to unique parameters 0.8 (urban), 0.5 (rural), taken from Grill et al (2018). But these are not process-based. It appears that are from back calculation or calibration for the specific case of China and maybe not valid for other areas.

[Figure]

fraction (dimensionless) of contaminant load from untreated pathways that reaches a waterbody after processes of natural attenuation. For example, for baseline model applications (see Section 4), the direct discharge coefficient for urban populations [1] as set to 0.8 and for rural populations to 0.5, respectively, following Grill et al. (2018). The higher coefficient value for urban areas is due to the presence of impervious surfaces leading to more direct disposal of wastewater to nearby rivers and streams. The total input of contaminants from untreated pathways to each river reach is then calculated as:

$$L_{u,r} = \left( \left( \sum_{m}^{c} P_{urb,m} \times ddc_{urb} \right) + \left( \sum_{m}^{c} P_{rur,m} \times ddc_{rur} \times F_m \right) \right) \times L_{cap} \tag{2}$$

where $L_{u,r}$ is the total load of the contaminant arriving at reach $r$ from all untreated pixels $m$ inside the reach catchment c (g day$^{-1}$); $L_{cap}$ is the per capita load (excreted) of the contaminant (g cap$^{-1}$ day$^{-1}$); $P_{urb,m}$ and $P_{rur,m}$ are the total count of population (persons) following the untreated pathway from pixel $m$, in urban and rural areas, respectively; and $ddc_{urb}$ and $ddc_{rur}$ (dimensionless) are the direct discharge coefficients representing the proportion of contaminant load from untreated pathways that are discharged into the river reach $r$ from urban and rural areas respectively. $F_m$ (dimensionless) is a factor by which loads from rural populations are additionally reduced based on an inverse distance relationship that accounts for limited connectivity in areas that are further away from the river network, following the approach by Grill et al. (2018):

$$F_m = (D_{m,r} + 1)^{-2} \tag{3}$$

where $F_m$ (dimensionless) represents the fractional distance-based contribution factor for pixel $m$; and $D_{m,r}$ (kilometers) is the Euclidean distance between the pixel $m$ and the river reach $r$. This equation delivers fractional contribution values between 0 and 1, with 1 for locations closest to the river, 0.5 at a distance of 1 km, and continuously decreasing values as the distance increases. In contrast to the original method used in Grill et al. (2018), we refrained from normalizing the factor (i.e., by constraining $F_m$ to 0 at the furthest distance in each reach catchment), considering that contaminant contributions from any distance can reach the river system. Also, [3] e used Euclidean distances rather than distances along the surface hydrological flow path (as proposed by Grill et al. 2018) assuming that contaminants can also travel through soils and groundwater. We tested the sensitivity of the parameter settings by doubling and halving both the distance value and the exponent in Equation 3, finding that the resulting uncertainty ranges were below those of other model parameter settings.

**3.3 River and lake routing**

The mass transport in HydroFATE follows a 'plug-flow' approach (Pistocchi et al., 2010). That is, a 'plug' of substance mass is accumulated downstream as the sum of the input from the current and all upstream reaches flowing into the current reach (Grill et al., 2018):

$$L_{a,r} = \left( L_{t,r} + L_{u,r} + \sum_n L_n \right) \times d_{s,r} \times d_{l,r} \tag{4}$$

Number: 1    Author: Anonym    Subject: Highlight    Date: 14/08/2023 14:17:06
This is something that can be better estimated. Perhaps whith the residence time.

Number: 2    Author: Anonym    Subject: Highlight    Date: 24/08/2023 14:29:56
Why using this Fm parameter only for untreated rural effluents? Why not also for treated and untreated urban effluents? The residence time in the sewage system also leads to degradation depending on the distance.

Number: 3    Author: Anonym    Subject: Highlight    Date: 24/08/2023 14:43:08
The Euclidean distance must be better discussed, also keeping in mind that groundwater flow does not follow straight lines, but also follows flow directions trough positive gradients related to the local terrain geomorphology.

Moreovoer, since you state that the model is not very sensitive to this parameter, why did you add it on your model? This way, it seems that you add unnecessary complexity. For instance, the ddc parameters and their variability in your scenarios, should be already enough to account for the uncertainty.

[revised manuscript text omitted]

Number: 1    Author: Anonym    Subject: Highlight    Date: 24/08/2023 10:06:45

It seems that you are missing an important dataset of measurements

Wilkinson et al. (2020), https://doi.org/10.1073/pnas.2113947119.

Above, you cited them but their database of MECs seems not in yours.

Number: 2    Author: Anonym    Subject: Highlight    Date: 24/08/2023 14:56:22

Which is the rationale to assess the dominance of human use over vet. or industrial?
It seems (but I cannot see the MECs location in the database) that some of the adopted MECs are on a major rivers and basins which certainly have relevant veterinary use.

Number: 3    Author: Anonym    Subject: Highlight    Date: 24/08/2023 14:58:00

I do not understand how width and discharge would help you in locate the MECs points.

[Figure]

(including full substance removal in lakes) to represent low-end contaminant concentrations in the river network, and vice versa for Scenario 4. Additional scenarios (see Table S-3 in the supplementary material) were designed for the model performance evaluation to analyze the individual contributions of selected parameters and model configurations on the output, including a worst-case scenario assuming that no removal processes affect the contaminant load.

400 **Table 1. Scenarios designed to represent plausible parameter and model configuration settings to simulate the global distribution of SMX in rivers.Excretion fraction is the fraction of the consumed amount of SMX that is excreted after metabolism. Wastewater treatment removal efficiency is the percentage of SMX that is removed in treatment facilities (WWTPs or DWTS). For other parameters and configurations see text.**

| | Scenario | Parameter settings | | | | | Configuration settings | |
| --- | --- | --- | --- | --- | --- | --- | --- | --- |
| | | Excretion fraction | Wastewater treatment removal efficiency (%) | Direct discharge coefficient | | Instream decay constant $k$ (day$^{-1}$) | Lake removal | Discharge condition |
| | | | | Urban $ddc_{urb}$ | Rural $ddc_{rur}$ | | | |
| 1 | *Baseline, average-flow* | 0.2 | 49 | 0.8 | 0.5 | 0.13 | CSTR removal | Average-flow |
| 2 | *Baseline, low-flow* | 0.2 | 49 | 0.8 | 0.5 | 0.13 | CSTR removal | Low-flow |
| 3 | *Low-end case* | 0.1 | 73 | 0.2 | 0.2 | 2.88 | Full removal | Average-flow |
| 4 | *High-end case* | 0.3 | 2 | 0.9 | 0.9 | 0.03 | No removal | Low-flow |

**4.2.2 Exposure assessment**

405 The ratio of PEC to PNEC was used as an indicator to designate levels of SMX in the global river network that can lead to environmental health concerns. For that purpose, risk quotients, $RQ_{r,S}$ (dimensionless), were calculated for every river reach $r$ and every scenario configuration $S$ using the value of PNEC for SMX of 600 ng L$^{-1}$ and the calculated $PEC_{r,S}$ (ng L$^{-1}$) at every reach $r$ for the respective scenario $S$:

$$RQ_{r,S} = \frac{PEC_{r,S}}{PNEC} \tag{8}$$

410 In instances where the risk quotient is greater than or equal to 1 ($RQ_{r,S} \geq 1$), it is assumed that this exposure level can cause negative environmental impacts (Archundia et al., 2018; Hernando et al., 2006).

**4.2.3 Performance evaluation**

The performance of the model was evaluated by comparing reported MECs of SMX (see Section 4.1.3) and PECs calculated using HydroFATE at the coinciding river reaches. The goodness-of-fit indicators used to quantify model performance included 415 the normalized root mean square error (NRMSE), the percentage of bias (PBIAS), the Nash-Sutcliff efficiency coefficient (NSE; Nash & Sutcliffe, 1970), and the Kling-Gupta efficiency coefficient (KGE; Gupta et al., 2009). Besides the baseline calculations of Scenario 1, Scenarios 3 and 4 were evaluated as they benchmark plausible low-end and high-end variations of parameter and configuration settings, and Scenarios 5-7 (including 14 sub-scenarios, see Section S.3 in supplementary information) were developed to specifically test the uncertainty ranges introduced by individual parameter and configuration 420 settings.

Number: 1    Author: Anonym    Subject: Highlight    Date: 24/08/2023 11:35:15
In my opinion, the low and high-end scenario may have be better designed.

Why did you use the average flow and not the high flow?

Why you did not include all the scenarios you have in  table S3 for the validation, i.e., in the Fig 5e?

Number: 2    Author: Anonym    Subject: Highlight    Date: 24/08/2023 15:04:51
All parameters should be justified. For instance, above you discussed the WWTPs removal variability in literature being 2% (min), 49% (ave), 73 (max). What about the other values? I do not see on what the max min and ave values are based on.
Because are crucial parameters , I also ask to discuss all the values and their max min values in the text.

PArticularly, the ddc parameters variability is not clear. 0.8 and 0.5 are based on the Grill et al (2018) which I already discussed on a comment above. What about their max min values? What are based on?

I would ask to put the reference of all parameters in the table (or in the table of supp material).

Number: 3    Author: Anonym    Subject: Highlight    Date: 24/08/2023 09:46:23
why not using high-flow to account for low-end case?

[revised manuscript text omitted]

Number: 1     Author: Anonym     Subject: Highlight   Date: 28/08/2023 17:43:32
This is a nice way to explicit the river and lake  ecosystem services of degrading the SMX.

You can also do the same not only for the RQ>1 but also for the raw results. And you can also separate the effects of WWTP and ecosystem services therefore express the importance of technology and ecosystems service separately and express them in terms of % of reduction or load reduction.

Number: 2     Author: Anonym     Subject: Highlight   Date: 28/08/2023 16:57:26
Is "total" really necessary here? The table is just about selected countries.

[Figure]

[Figure]

to be particularly important in rivers in South Africa, United States, Mexico, and China. For instance, without lake removal,
500   there would be an increase of 51% of rivers in South Africa falling in the risk category at low-flow conditions.

Finally, to demonstrate the utility of a contaminant fate model operating at high spatial resolution, Figure 4 depicts the risk distribution under low-flow conditions (Scenario 2) for the region of Southeast Asia, including the two countries (India and Pakistan) with the longest total length of rivers in which SMX concentrations exceed PNEC. The high spatial resolution permits the detection of local increases in risk immediately downstream of individual WWTPs, which then can diminish along the
505   flow paths once inflowing tributaries cause dilution effects. Model results also reveal the exposure of individual rivers receiving contaminant discharge without any treatment (i.e., areas without any black dots but presenting a high density of rivers at risk).

[Figure]

**Figure 4. Estimated risk quotient for sulfamethoxazole (SMX) using HydroFATE in parts of Southeast Asia, with a focus on India**
510   **and Pakistan. The risk quotient was calculated using Scenario 2 (see Table 1 for scenario settings). The black dots represent the location of wastewater treatment plants (WWTPs) in the database HydroWASTE. For visual clarity, only rivers exceeding a long-term average flow of 0.1 m$^3$ s$^{-1}$ are shown on the map.**

[Figure]

**4.4 Performance evaluation**

Modelling results were evaluated by comparing predicted SMX concentrations with available measurements in river reaches

515 across the world using 227 MECs with values above the detection threshold. Figures 5(a) to (d) show an analysis of results for baseline Scenario 1 (average-flow conditions; see Table 1 for scenario settings), ratified by certain characteristics of the measurements. The different colors of points in the overall scatter plot shown in Figure 5(a) illustrate the global distribution of measurements. The African continent presents the highest SMX concentrations (both measured and predicted) and the predicted concentrations in Asia, Europe and Central America are in their majority below reported measured concentrations.

520 These results confirm that model predictions for Scenario 1 are generally reasonable, with 79% of the predicted values being within one order of magnitude of the measured concentrations reported in literature (Johnson et al., 2008, Oldenkamp et al., 2018).

Figure 5(b) shows the different model performance indicators when comparing all 227 MECs against estimates derived for Scenario 1, reflecting overall fair results yet with a clear bias towards modelled concentrations being lower than reported

525 measurements. Since there is a possibility that some of the MECs were measured during low-flow conditions (Section 4.1.3), the comparison with results using Scenario 1 (representing average-flow conditions) might not be appropriate. To explore this issue further, error bars were added into Figure 5(b), where the ends of the error bars (extending only leftwards) show a recalculated MEC for average discharge conditions when assuming that the original concentration of SMX was measured during low-flow conditions. This analysis demonstrates that cases in which predicted values were too low could, in part, be

530 explained by uncertainties within the measurements rather than errors in the model predictions.

Number: 1     Author: Anonym     Subject: Highlight   Date: 28/08/2023 17:03:40

Firstly, this requirement was not explained in methodology section.
Secondly, picking only MEC values above detection limit is a subjective choice that hides possible strength and weakness of the model. For instance, if MEC is below detection, but PEC is above detection, or vice-versa, the model should have lower performance. MECs below detection are still valuable data to validate. against.

Number: 2     Author: Anonym     Subject: Highlight   Date: 28/08/2023 17:05:01

Stratified or classified?

Number: 3     Author: Anonym     Subject: Highlight   Date: 28/08/2023 17:39:04

You have already included the uncertainty in model prediction due to discharge condition into your PECs. If you include it again in MECs, it means that you are considering this uncertainty twice, which is not correct.

You may introduce the uncertainty (or variability) in MECs if you have repeated measurements at the same point.

Otherwise, if you find it necessary and if you have measured discharge for some of the MECs, you can compare loads instead of concentrations.

[Figure]

**Figure 5. Evaluation of the SMX concentrations estimated using HydroFATE for baseline Scenario 1 (average-flow conditions; see Table 1 for scenario settings)ndividual points represent comparisons between Measured (MEC) and Predicted (PEC) Environmental Concentrations in the same river reach. The black dashed line represents the 1:1 line and the gray dashed lines represent the error**
535 **line corresponding to one order of magnitude. Panel (a) shows a general scatter plot between MECs and PECs and indicates their spatial distribution (with different point colors representing the geographic regions). Panel (b) shows uncertainties of MECs regarding river discharge where the ends of the error bars represent a modified MEC assuming the original was measured during low-flow conditions; n is the number of records; NRMSE is the normalized root mean square error; PBIAS is the percent bias; NSE is the Nash-Sutcliffe efficiency; and KGE is the Kling-Gupta efficiency. Panel (c) indicates the annual river discharge at the river**
540 **reach of the MEC in different bubble sizes; and panel (d) indicates the urban extent upstream of the river reach of the MEC in different bubble sizes; the red dashed lines represent the threshold for the PNEC. Panel (e) shows uncertainties of PECs where the error bars represent the range of resulting PECs using the parameter and configuration settings of Scenarios 3 and 4. Error bars that extend below 0.001 ng L$^{-1}$ may include predicted zero concentrations.**

Number: 1   Author: Anonym   Subject: Highlight   Date: 28/08/2023 17:45:15
I would appreciate a discussion on these quality parameters values (NRMSE, NSE, PBIAS, KGE). Are they expressing a good or bad prediction performance of your model?
How do they compare with other similar large scale models performance?

Number: 2   Author: Anonym   Subject: Highlight   Date: 28/08/2023 17:23:29
Is this statistic necessary?

Number: 3   Author: Anonym   Subject: Highlight   Date: 28/08/2023 17:23:40
Is this statistic necessary?

Number: 4   Author: Anonym   Subject: Highlight   Date: 28/08/2023 17:14:12
As explained above, I  do not think that this is possible.

[Figure]

Figure 5(c) shows the same points with bubbles sized according to average discharge at the measurement location and Figure
545  5(d) shows bubbles sized according to upstream urban extents. Most measurement locations for which model predictions of
concentrations were too high are located downstream of urban populations on rivers with low discharge, which is a challenging
combination to model; that is, urban streams can be heavily modified by anthropogenic activities that influence their flow
quantities and water quality, such as channelization, dams, and sewers. Besides, if the urban population is not served by
WWTPs, the predictions were based on the assumption of a constant direct discharge coefficient, which in all probability is
550  variable in reality. Despite these uncertainties, only two measurements were predicted erroneously above the PNEC threshold,
resulting in a risk quotient that is falsely predicted to be above 1. On the other hand, 17 measurements were erroneously
predicted to be below the PNEC threshold, which is in accordance with the overall conservative approach and scenario
configuration used in this case study.

Finally, Figure 5(e) shows substantial uncertainties in PEC calculations (i.e., extent of error bars) when using the parameter
555  and configuration settings of Scenarios 3 and 4, representing low-end and high-end simulations that were within plausible
ranges. Taking these uncertainties into account, 78% of all MECs fell within the range of error bars of HydroFATE; i.e., they
were reproducible by the model within at least one of the chosen parameter and configuration setting.

In addition to the 227 MECs,
134 measurements were classified as 'not detected' or 'not quantified.' To evaluate these cases,
PECs at the same locations were verified to determine if they were correctly predicted to be below the detection or
560  quantification limit (LOD or LOQ, respectively), depending on the limit reported by the study. The rate of success was 60%,
and if allowing an error of one order of magnitude, the success rate increased to 93%.

In addition to the performance evaluation presented in Figure 5, three additional scenarios (including 14 sub-scenarios) were
analyzed to specifically test the uncertainty ranges introduced by individual parameter and configuration settings (see Section
S.3 in supplementary information for details). The results indicate that the model reacts sensitive yet within reasonable
565  boundaries to the permutations of individual parameter settings. Of particular importance are the settings related to substance
removal simulations in the model which depend on local characteristics and on the dominating pathway of the contaminant.

**5 Discussion**

**5.1 HydroFATE: strengths and limitations**

In this study, the first global application of the contaminant fate model (CFM) HydroFATE was presented, building upon
570  previous stages of the model that were used to assess the distribution of pharmaceuticals at the regional scale in Canada, China
and India. One of the main characteristics that distinguishes HydroFATE from other global CFMs, besides its high spatial
resolution, is that contaminant pathways are differentiated based on whether the wastewater undergoes treatment or not and,
if so, at what treatment level. Contaminants generated by populations connected to wastewater facilities are partially removed
by treatment processes, whereas contaminants generated by populations not connected to any wastewater system are assumed
575  to undergo natural attenuation processes, which in the case of rural populations also depend on how distant they are from any

Number: 1     Author: Anonym     Subject: Highlight   Date: 28/08/2023 17:44:24
Therefore, you used also MECs below limit of detection for validation. Why did you write the opposite at paragraph 515?
And why you did not introduced these 134 values in methodology section?

Number: 2     Author: Anonym     Subject: Highlight   Date: 28/08/2023 18:21:56
This is misleading because at paragraph 355 you wrote that treatment level (primary, secondary…) is not taken into account here for SMX.
Please, clarify this.

[revised manuscript text omitted]

---

## Author Comment (AC1)

**Response to Reviewer 1**

**R1-C1**

*Regarding the lower calculated concentrations in river, the authors have indicated that river discharge may be higher than observed and veterinary and industrial are not considered. I agree on this point. But on the other hand, it should be mentioned that there are factors that further decrease the river concentration. Within this paper, there is no mention of load reduction before entering the wastewater treatment plant. If the direct discharge coefficient is considered as the inflow from the conduit to the river in urban untreated area, I believe that something similar may be happen in the sewer pipes. Taking this into account will lead to a decrease of river concentration. In addition to this, advanced wastewater treatment plants could lead to further load reductions. It would be desirable to mention these points in the discussion or as future research topics.*

A: We agree with the reviewer that sewer leakage, environmental decay and other losses before the contaminant load enters wastewater treatment plants could substantially reduce the river concentration. I.e., if we added these processes to our model, our simulations would lead to even lower calculated concentrations, hence our bias would be further amplified. That said, we added a new paragraph after line 596 to discuss these additional uncertainties:

*"The efficiency of a WWTP to remove a specific contaminant is also a complex process that depends on characteristics of the individual facilities and local conditions that are not represented in the global HydroWASTE database. Furthermore, processes not simulated by HydroFATE may have an impact on contaminant loads entering surface waters. For example, depending on how far a household is located from the facility, decay processes in sewers can act on the load on its way to the WWTP. In addition, sewer lines that are poorly maintained may result in wastewater leakages into the ground, further reducing the load of contaminant before it reaches the WWTP."*

Also, we changed the statement at line 649:

*"As it is not possible to isolate the contribution from domestic sources in the MECs, this uncertainty in both PECs and MECs could explain a portion of the high negative bias found in the evaluation."*

We also agree that WWTPs with an advanced level of treatment would typically lead to higher load reductions, thus HydroFATE has the capability to differentiate efficiencies from different levels of treatment. Our decision to use the same efficiency for all treatment levels in our test case study was due to the lack of data on this parameter for the chosen substance sulfamethoxazole. This uncertainty is discussed in lines 635-640, but we added a new statement at line 640:

*"This could lead to reduced river concentrations, since secondary and advanced treatment processes are expected to result in higher removal efficiencies."*

Minor comments:

**R1-C2**

***It would be good to indicate the 10 km on line 147, if there is any reasoning behind it.***

A: According to the methodology used to estimate the outfall location of the wastewater discharge location in the global WWTP database HydroWASTE, 10 km is the maximum distance between the actual facility and the outfall location and it is a described uncertainty in HydroWASTE. The distance was selected based on a statistical determination process using a subset of WWTPs and remote sensing imagery for manual verification (see Ehalt Macedo et al., 2022). To clarify this in the manuscript we added in line 148:

*"..., given the locational uncertainties in HydroWASTE of up to 10 km (Ehalt Macedo et al., 2022),..."*

**R1-C3**

***Isn't ds,r in line 321 a mistake for dl,r?***

A: We thank the reviewer for spotting this typo, though the error is actually in line 322 rather than in the equation. We corrected line 322 to "...$d_{l,r}$ (dimensionless) is the lake decay factor ...".

**R1-C4**

***Check line 332 for a reference error.***

A: The reference error in line 332 has been reviewed and corrected.

**R1-C5**

***There is a spelling error (individual) in 533 in Fig. 5.***

A: The spelling error "individual" in the caption of Figure 5 (line 533) has been fixed.

**References**

Ehalt Macedo, H., Lehner, B., Nicell, J., Grill, G., Li, J., Limtong, A., and Shakya, R.: Distribution and characteristics of wastewater treatment plants within the global river network, Earth Syst. Sci. Data, 14, 559-577, doi: 10.5194/essd-14-559-2022, 2022.

---

## Author Comment (AC2)

**Response to Reviewer 2**

**Critical:**

**R2-C1**

*According to GMD code and data availability policy: "Where the authors cannot, for reasons beyond their control, publicly archive part or all of the code and data associated with a paper, they must clearly state the restrictions" (https://doi.org/10.5194/gmd-12-2215-2019). In the supplement python file, HydroFATE_v09.py, the imported library arcpy is not publicly available. Please state the restrictions in the code and data availability section. For example: "A license of the software provided by ... is required to run the provided scripts."*

A: We acknowledge the important issue about the arcpy library not being publicly available. We updated the code and data availability section to clearly state this restriction at line 667:

"*A license of the software ArcGIS Pro provided by ESRI is required to run the provided scripts.*"

**R2-C2**

*The method presented in Section S.1 of the supplementary material is crucial to reproducing the modeling result of HydroFATE v1.0. In essence, Section S.1 details the generation algorithm of the WWTP service area. Section S.1 should be moved to the main manuscript, perhaps in the Appendix.*

A: We agree that the generation algorithm of the WWTP service area (Section S.1) is crucial to reproducing our modeling results (we originally had placed the descriptions into the SI only for space considerations). We now moved this section to the main manuscript as an appendix, to make it more accessible to readers.

**R2-C3**

*Follow up on the previous comment on the generation of the WWTP service area. Line 242, which is in Section 3.1, reported a "successive trial-and-error approach in which intermediate results were mapped, visually inspected for plausibility, and statistically tested to verify whether they led to further improvements." I understand the difficulty of the service area generation, and it is acceptable to inspect the model visually. However, the realization of the WWTP service area is a critical part of the HydroFATE model. Hence, it is within the scope of the GMD code and data availability policy. I urge the authors to provide the scripts you use to generate the*

*service area since I cannot find them in HydroFATE_v09.py. (Please review Section 3.2 of the GMD editorial: https://doi.org/10.5194/gmd-12-2215-2019.) Furthermore, the sensitivity analysis in Section S.3 does not provide the sensitivity of different realizations of the WWTP service area. Therefore, it is necessary to make the service area generation process more transparent by providing the script.*

A: We thank the reviewer for this comment. We understand the concern about the transparency of the WWTP service area generation process. To address this, in the revised version of the paper we will provide the script used to generate the WWTP service areas, as suggested in Section 3.2 of the GMD editorial. This will enhance the reproducibility and transparency of our work. As such, we added the following sentence at line 659:

*"The code for the WWTP delineation and its output is also available under at the same URL."*

**Suggestions:**

**R2-C4**

*Figure 2 provides an example of the generated WWTP service area used in this study. It would be helpful to show examples of intermediate iteration results and explain why this iteration is rejected or accepted so the readers can have a gist of the visually inspected acceptance criteria.*

A: We have thoroughly considered this interesting comment. While we understand the value of a more comprehensive representation of the iterative steps involved, we struggled in finding representative in-between stages that are easy to explain yet illustrative and representative for the broad range of issues that we encountered. We felt that either the figure and explanations would get overly complex, or the simplifications may cause further confusion for the reader. To stay within reasonable limitations in terms of space and detail in this publication, we therefore hope that including the script and the output in the revised paper (see previous comment and answer) will be sufficient to help readers who have an interest in applying our methodology in their own work, or to experiment with modifications to the algorithms to study the associated effects.

**R2-C5**

*In equation (1), the removal efficiencies of wastewater treatment plants (WWTP), $e_{WWTP, j}$, contain three treatment levels, which are primary, secondary, and advanced. However, the parameters of the case study do not include such complexity. Only one removal efficiency is*

*used in each scenario. During the review, I have always considered WWTP treatment levels spatially varying, and they can be sensitive to the modeling result. I find the presentation of Figure 2 a bit misleading. Therefore, I suggest adding another figure in Section 4, which is based on Figure 2, but the WWTP areas are only color-coded once to represent the model better.*

A: We understand and appreciate this concern. Our goal is not to mislead the reader by presenting different WWTP types in Figure 2. But we would like to emphasize that our case study, presented in Section 4, serves only as a test application of the model. In that case, regrettably, certain parameters essential for the selected substance were unavailable for inclusion in the study and some aspects of the analysis presented in Section 4 may not fully capture the complete scope of the model's capabilities. Therefore, we believe that adding another, simplified figure would not do justice to presenting the comprehensive and accurate features of the model. The model has explicitly been developed to be able to distinguish different WWTP types, it is only in the test study where we could not implement this distinction. We therefore hope that it is acceptable to leave the presentation of Section 4 as is. We would also like to point to Figure 4 which presents the WWTPs used in India, all visualized without their specific type.

**Minor:**

**R1-C6**

*The unit in Figure 4 should be presented as $m^3 \ s^{-1}$.*

A: As requested, we corrected the units in the caption of Figure 4 to be presented as $m^3 \ s^{-1}$.

**R1-C7**

*Fix the unit in Figure S-3: $ng \ L^{-1}$.*

A: We fixed the units in the caption of Figure S-3 to be $ng \ L^{-1}$.

**R1-C8**

*Fix the reference error in Table S-5 of the supplement.*

A: We corrected the reference error in Table S-5 of the supplementary materials.

---

## Author Comment (AC3)

**Response to Francesco Bregoli (Reviewer 3)**

**General comments**

**Methodology**

**R3-C1**

*The mechanism of water/soil partitioning (or absorption to soil) is complex. Here, the choice of their relative parameters is not process-based. It appears that are from back calculations or calibrations for the specific case of China (Grill et al. 2018) and may be not valid for other areas of the world. I understand that this is difficult for such global scale. But this needs to be better discussed.*

A: We appreciate the thoughtful consideration of the complexities involved in the mechanism of water/soil partitioning and the associated parameter choices. We fully agree that the relative parameters used for the calculation of the direct discharge coefficient may not be universally applicable, as they were derived from a specific case study in China (Grill et al. 2018). While it would certainly be preferable to include region-specific parameter settings, the lack of data to support refinement of such parameters throughout the globe represents, at present, a considerable challenge. In recognition of this, we acknowledge that these uncertainties can substantially impact the outcomes, especially in regions with limited treatment infrastructure (see original manuscript lines 596-609). Nevertheless, we believe that the introduction of the direct discharge coefficient into the model is a crucial design feature and improvement of the model, since in previous large-scale contaminant fate models untreated pathways were not considered at all.

While recognizing the limitations, we would like to reiterate that our study is, in essence, a test of the model on a global scale. The limitations arising from unknown processes and measurement uncertainties are integral to this pioneering effort and set the stage for further refinement, development and future application of the model.

In order to further discuss this issue, as requested, and to reiterate the uncertainty while expressing the importance of the direct discharge coefficient, we added the following text after line 283:

*"While these methods are simplistic in comparison to the real soil processes, no previous large-scale model considers untreated pathways as sources of contaminants, which can be substantial in regions with limited treatment infrastructure."*

And we also revised and expanded the text at line 601:

*"In fact, Grill et al. (2018) found in a sensitivity analysis for China that the setting of the direct discharge coefficient in rural areas represented the main source of model uncertainties. However, while the simplification of soil-related processes and the determination of their spatially heterogenous parameter settings remain a major challenge and likely source of error, in particular in areas dominated by untreated pathways, these simulations are critically important to be implemented in the model design. For example, in the presented case study, the untreated pathways contributed an estimated 72% of the global emission of sulfamethoxazole, demonstrating their decisive role. Despite the large uncertainties, the baseline scenario was able to reproduce field measurements reasonably well, especially considering the large range of possible values for the direct discharge coefficients (see Table 1). "*

**R3-C2**

***On-soil and groundwater contaminant degradation are here accounted with a degradation parameter based on the Euclidean distance from hypothetical source point and closer stream. But, groundwater flow does not necessary follow straight lines, but also follows flow directions trough positive gradients related to the local terrain geomorphology. Also, aquifers increase residence time, and therefore degradation. Because you state that the model is not very sensitive to this parameter, why did you add it on your model? This way, it seems that you add unnecessary complexity.***

A: We agree that the hydrological flow path may be closer to reality than Euclidean distance in catchments that are governed by clear topography. But considering the uncertainty inherent in establishing hydrological flow paths at a global scale, including in challenging areas such as floodplains, we decided to simplify the process (it should be noted that in the original model implementation the flow path concept was used, see Grill et al. 2018). However, while the model was not 'very' sensitive to this process, we believe that accounting for the relative reduction in contaminant loading into rivers due to the location of population settlements and for variability in drainage density is an important model feature. For example, it may lead to improved simulations for particular regions which may not have been detected in the global sensitivity analysis. By including it in the model design, we hope it can be tested more rigorously in the future.

**SMX case study and validation**

**R3-C3**

*SMX has both human and veterinary use. Therefore, it is complex to account only for human uses when validating the model. In the validation, you attempted to focus only on catchments were human use dominates. However, at global scale and with such big basins, this is very difficult.*

A: We fully agree with this comment and acknowledge that to exclude other sources of antibiotics in measurements is virtually impossible, especially for large rivers. Our attempt was only to avoid measurements that explicitly reported in their description that other sources were included, to avoid known cases where measurements and predictions are by default expected to be incomparable. We also noted in the original manuscript that other sources are a possible cause of the negative bias observed in our model results, as discussed in lines 648 to 652. To clarify the methodology used, we changed the explanation regarding the selection of measurements in lines 370 to 374:

*"In order to be selected for inclusion as a MEC in the model evaluation, the literature source must have reported the specific location (i.e., in the form of coordinates, river names, or river intersections) of the measurements. In addition, we discarded any MEC where the literature source explicitly mentioned that the dominant use of antibiotics in the catchment feeding the river was associated with veterinary or industrial activities, since the current version of HydroFATE is not adapted to account for these sources."*

**R3-C4**

*You defined several scenarios by changing relevant parameters. However, all parameters choice should be justified. For instance, you discussed the WWTPs removal variability in literature being 2% (min), 49% (ave), 73 (max) based on literature values. What about the other parameters in the different scenarios?*

A: The excretion fractions and the instream decay constants are also based on literature sources, as described in detail in Section 4.1.1, as stated in the manuscript (line 392). The direct discharge coefficients of Scenarios 1 and 2 are taken from the existing study for China (Grill et al. 2018; see line 282)—we added this reference to line 392 (*"... and in Grill et al. (2018) for the direct discharge coefficients."*).

This leaves only the direct discharge coefficients for Scenarios 3 and 4 unexplained. Our goal with Scenarios 3 and 4 was to present minimum and maximum predicted concentrations within reasonable parameter variations. Since the direct discharge coefficient is a highly uncertain parameter developed based on a previous study (Grill et al. 2018), it has almost arbitrary variability. For this reason, we selected the values of 0.2 and 0.9 as plausible values that are close to the extremes of 0 and 1, which were analysed in the supplementary scenarios. To clarify this, we added an explanation to line 397:

*"In the absence of relevant literature values, plausible boundaries for the direct discharge coefficients of Scenarios 3 and 4 were set slightly above 0 (with 0 representing complete decay along untreated pathways) and below 1 (representing no decay along untreated pathways)."*

Finally, we rephrased the last statement in the Table caption to add clarity:

*"For parameter settings and configurations see text."*

**R3-C5**

***High flow conditions, although favourable for higher dilution, is not considered as extreme low-end scenario.***

A: As also indicated in the previous response above, Scenarios 3 and 4 are not supposed to represent "extreme" conditions, but rather reasonable/plausible conditions within the parameters' uncertainties. To avoid misunderstanding and to clarify that we are not using high-flow conditions, we changed the names of Scenarios 3 and 4 to *"Low-end case, average flow"* and *"High-end case, low flow"*.

**R3-C6**

***The choice of using or not MECs below detection limit for validation is contradicted a couple of time in the MS. I would describe it better and univocally in the methodology section. In my opinion, MECs below detection limit are still important for model validation.***

A: We apologize for not being clear when introducing the methodology associated with the selection of MECs. In the revised manuscript, we moved the following statement from line 558 to line 416:

*"In addition to the 227 MECs, 134 measurements were classified as 'not detected' or 'not quantified.' To evaluate these cases, PECs at the same locations were verified to determine if they were correctly predicted to be below the detection or quantification limit (LOD or LOQ, respectively), depending on the limit reported by the study."*

Furthermore, line 560 was corrected to:

*"For PECs at the same locations as MECs that were reported to be below detection, 60% were correctly predicted to have concentrations that fall below the detection limit. If allowing an error of one order of magnitude, the success rate increased to 93%."*

We also corrected the statement appearing at line 515:

*"Modelling results were evaluated by comparing predicted SMX concentrations with available measurements in river reaches across the world using 227 MECs with values above the detection threshold and 134 measurements below detection."*

**R3-C7**

***You accounted the uncertainty in model prediction due to discharge condition into your PECs. If you include it again in MECs, it means that you are considering this uncertainty twice, which is not correct.***

A: After due reflection and consideration, we respectfully disagree with this assertion. It is important to note that the uncertainties depicted in panel b of Figure 5 pertain to MEC uncertainties, rather than being indicative of model uncertainties. The purpose of including this graph was to illustrate how unknown characteristics specific to MECs might contribute to disparities between MECs and PECs at the same location. Such disparities should not be categorized as model uncertainties.

Nevertheless, in order to provide greater clarity on this matter, we have relocated panel b from Figure 5 (Section 4.4) to the Discussion section (Section 5.2), along with its accompanying explanation. This adjustment aims to facilitate a clearer differentiation between uncertainties related to model predictions and those associated with the evaluation data.

**Discussion**

**R3-C8**

*I would appreciate a deeper discussion on the quality prediction (i.e. on NRMSE, NSE, PBIAS, KGE parameters). Are they expressing a good or bad prediction performance of your model? How do they compare with other similar large scale models performance? Is it an acceptable performance for predicting contaminants at global scale?*

A: We agree that the goodness-of-fit indicators should be further discussed and, as such, we revised the text appearing at line 642:

*"The results showed an overall reasonable predictive capability with the goodness-of-fit indicators NSE and KGE above 0.6, and with 79% of PECs being within one order of magnitude of reported MECs. This was despite the inherent uncertainties associated with assumptions made in the development of the model and those associated with estimates of the various model parameters and input datasets. Unfortunately, the lack of specificity of field measurements, for which literature sources generally do not provide enough information on the precise locations of measurements nor river discharge conditions, does not allow for a conclusive evaluation of the model under different modelling scenarios. It is noted that other global water quality models, which also simulate substance loads and concentrations, have reported similar values of NSE between 0.4 and 0.71 (Font et al., 2019; Harrison et al., 2019). However, a more detailed comparison between results from these models and HydroFATE is difficult as different substances and spatial resolutions were applied."*

**Extra comments extracted from the annotated pdf:**

*It seems that you are missing an important dataset of measurements, Wilkinson et al. (2020), [https://doi.org/10.1073/pnas.2113947119](https://doi.org/10.1073/pnas.2113947119). Above, you cited them but their database of MECs seems not in yours.*

A: The dataset by Wilkinson et al. was only released in 2022, i.e., after our data collection and implementation phase had been completed. Adding these data to our assessment would represent a major amount of work (and thus substantial delay in publishing this work) as all measurements would need to be georeferenced to our river network in a quality-controlled way before being assessed. Hence, we refrained from utilizing these new data at this stage. However, we do understand the value of these additional data and plan to use them in the future.

*I do not understand how width and discharge would help you in locate the MECs points.*

A: The river network does not always match the exact location in reality, and the coordinates of MECs are not always precise. Therefore, river characteristics such as discharge and width can help differentiate between different rivers. For example, at a river confluence the MEC may be recorded to fall between the main river and a small tributary. If discharge or width of the measurement location were known, the correct river could be more easily determined.

***Why not using high-flow to account for low-end case?***

A: High flows as long-term monthly averages are not as meaningful for risk assessments. Nonetheless, we changed the scenario denomination and hope it is clearer now (see Response R3-C5).

**References**

Grill, G., Li, J., Khan, U., Zhong, Y., Lehner, B., Nicell, J., and Ariwi, J.: Estimating the eco-toxicological risk of estrogens in China's rivers using a high-resolution contaminant fate model, Water Research, 145, 707-720, doi: 10.1016/j.watres.2018.08.053, 2018.

Wilkinson, J. L., Boxall, A. B. A., Kolpin, D. W., Leung, K. M. Y., Lai, R. W. S., Galbán-Malagón, C., Adell, A. D., Mondon, J., Metian, M., Marchant, R. A., Bouzas-Monroy, A., Cuni-Sanchez, A., Coors, A., Carriquiriborde, P., Rojo, M., Gordon, C., Cara, M., Moermond, M., Luarte, T., . . . Teta, C. Pharmaceutical pollution of the world's rivers. Proceedings of the National Academy of Sciences, 119(8), e2113947119. doi:10.1073/pnas.2113947119, 2022.